

# Terrain is a stronger predictor of peat depth than airborne radiometrics in Norwegian landscapes

Julien Vollering[1], Naomi Gatis[2], Mette Kusk Gillespie[1], Karl-Kristian Muggerud[1], Sigurd Daniel Nerhus[1], Knut Rydgren[1], and Mikko Sparf[1]

[1]Department of Civil Engineering and Environmental Sciences, Western Norway University of Applied Sciences, Norway
[2]Department of Geography, University of Exeter, United Kingdom

**Correspondence:** Julien Vollering (julien.vollering@hvl.no)

**Abstract.** Peatlands are Earth's most carbon-dense terrestrial ecosystems and their carbon density varies with the depth of the peat layer. Accurate mapping of peat depth is crucial for carbon accounting and land management, yet existing maps lack the resolution and accuracy needed for these applications. This study evaluates whether digital soil mapping using remotely sensed data can improve existing maps of peat depth in western and southeastern Norway. Specifically, we assessed the predictive value of LiDAR-derived terrain variables and airborne radiometric data across two, >10 km² sites. We measured peat depth by probing and ground-penetrating radar at 372 and 1878 locations at the two sites, respectively. Then we trained Random Forest models using radiometric and terrain variables, plus the national map of peat depth, to predict peat depth at 10 m resolution. The two best models achieved mean absolute errors of 60 and 56 cm, explaining one-third of the variation in peat depth. Terrain variables were better predictors than radiometric variables, with elevation and valley bottom flatness showing the strongest relationships to depth. Radiometric variables showed inconsistent predictive value – improving performance at one site while degrading it at the other. The accuracy of the national map of peat depth did not measure up to any of our remote sensing models, even though it was calibrated to the same data. Still, weak relationships with remotely sensed variables made peat depth hard to predict overall. Based on these findings, we conclude that digital soil mapping can improve existing, broad-scale maps of peat depth in Norway, but highly localized carbon stock assessments are best made from field measurements. Furthermore, the inability of models to identify peat presence outside known peatlands highlights the need for integrated mapping of peat lateral extent and depth. Together, these pathways promise more accurate landscape-scale carbon stock assessments and better-informed land management policies.

## 1 Introduction

Peat soils are a terrestrial carbon stock of global importance. They store 450-650 Gt of carbon, or about 30 % of global soil carbon, despite covering only 2–3 % of Earth's land (Xu et al., 2018; Friedlingstein et al., 2020; UNEP, 2022). Peatlands (areas with peat soil) are more carbon dense per square meter than any other ecosystem (Temmink et al., 2022). This makes them crucial to climate change mitigation. Intact peatlands sequester carbon and overall produce a negative temperature forcing





(Joosten et al., 2016). When disturbed, often by conversion to another land use, they can produce large greenhouse gas emissions (Ma et al., 2022).

One of the keys to the areal density of peatland carbon stocks lies in the third dimension: their depth. Peat soils range from zero to over ten meters deep (Widyastuti et al., 2024), so deep peats comprise a large volume. Their depth results from the accumulation of organic matter over thousands of years (Loisel et al., 2014; Joosten et al., 2016). In the anoxic and acidic conditions created by a high water table, plant material decay is slightly outweighed by new growth, and the surplus carbon is laid down as peat. There it remains sequestered as long as the water table stays high and prevents oxidative peat decomposition.

Peatlands are most common at high latitudes, and in regions with high cover they are frequently converted to human land use (UNEP, 2022). They are attractive for agriculture and forestry because they are flat, treeless, and have developed soils. However, other land uses also displace peatlands. In Norway – where flat lowland is relatively scarce but upward of 9 % of the land area is peatland (Bryn et al., 2018; Bakkestuen et al., 2023) – lawmakers have restricted peatland afforestation and cultivation in recent decades. Since then, a larger proportion of peatland loss is driven by construction (Flaget et al., 2024).

The spatial distribution of peat depth is often overlooked in land use planning and carbon accounting because peat depth is not mapped with sufficient coverage, resolution, or accuracy (Beilman et al., 2008; Hastie et al., 2022; UNEP, 2022). Maps are crucial because they link high-level targets to specific management decisions, unlike spatially aggregated estimates (OECD, 2022). For example, Norwegian regulations restrict conversion of peatland to farmland more strictly where peat depth exceeds 1 m, and the distribution of peat depth on a farmer's property determines whether they are allowed to convert peatland (Forskrift 40 om nydyrking, 1997, § 5a). Maps of peat depth also make it possible to quantify the effect of specific management decisions and thereby understand how local outcomes contribute to regional and national outcomes (OECD, 2022).

    Measuring peat depth on the ground by probing or ground-penetrating radar is straightforward, and a field survey can map a small area at low cost. However, surveying large areas is impractical when depth varies widely over short distances (e.g. 10 m) – as in many peatlands (Torppa, 2011; Proulx-McInnis et al., 2013; Henrion et al., 2024). A complementary approach from 45 soil science is digital soil mapping (DSM). DSM scales up field measurements from a set of locations to a wider area, by relating the measured values to other variables mapped over the area of interest. This approach has grown in importance with the availability of remotely sensed data and the advancement of methods for identifying patterns, especially through machine learning (Minasny et al., 2019; Wadoux et al., 2020).

    The crux for DSM of peat depth is the relationship between peat depth and each of the other, mapped predictors. For DSM 50 to work, these relationships must be strong and consistent over the area of interest. They can be purely correlative rather than causal, but mechanistic relationships are stronger and more consistent than non-causal ones. The *scorpan* framework for DSM suggests seven predictor classes to explore: other soil properties, climate, organisms, relief (topography), parent material, age, and spatial position (McBratney et al., 2003).

    For peat depth, the most practical and widespread *scorpan* factors are relief and spatial position. Spatial position is unique 55 because it is always known (with varying accuracy). However, the short range of spatial autocorrelation in peat depth limits its mapping value (Hengl et al., 2004). Relief, or topography, is widely and accurately mapped in digital terrain models (DTMs). For example, most of mainland Norway has at least one elevation measurement per square meter from airborne Light Detection





and Ranging (LiDAR) surveys. Moreover, mechanisms of peat formation are linked to topography. For example, a steep slope is unlikely to have a high water table, thereby limiting accumulation of peat.

Studies have consistently shown relationships between peat depth and topography, although the specific patterns vary. One of the most robust relationships is a negative correlation between depth and terrain slope (e.g. Holden and Connolly, 2011; Parry et al., 2012; Gatis et al., 2019). Peat depth also changes with elevation in many contexts, but with inconsistent directionality (e.g. Holden and Connolly, 2011; Parry et al., 2012; Rudiyanto et al., 2016, 2018; Koganti et al., 2023; Li et al., 2024). Also more complex derivations of topography, such as the Topographic Wetness Index and the Multi-Resolution Valley Bottom

Flatness index, have shown associations with peat depth in some studies (e.g. Rudiyanto et al., 2018; Koganti et al., 2023; Li et al., 2024). Some of the variation between studies in quantifying these relationships is undoubtedly attributable to issues of spatial scale – both the scaling of the topographic predictors and the resolution of the peat depth analysis.

     Another set of predictors related to peat depth are measurements of natural radioactivity from the ground surface. Gamma-ray spectrometry can survey the activity (decay counts per second) from radioactive isotopes in the earth's crust: potassium-

40, uranium-238, and thorium-232 (Reinhardt and Herrmann, 2019). These exist in bedrock (and mineral soils) and a peat overburden attenuates the radiation intensity at the surface. The degree of attenuation relates to properties of the overburden, especially composition and depth. Deep soil with high water content will attenuate radiation most (Beamish, 2013; Reinhardt and Herrmann, 2019). Thus, gamma-ray radiometrics integrate two *scorpan* factors: other soil properties and parent material (McBratney et al., 2003).

Although theory suggests that one meter of peat may fully attenuate radiometric signal (Beamish, 2013; Reinhardt and Herrmann, 2019), empirical investigations show that the association between peat depth and radioactivity can extend beyond the first meter (Keaney et al., 2013; Gatis et al., 2019; Koganti et al., 2023). Radiometric data are increasingly available over large areas, presenting a big opportunity for mapping peat depth, insofar as they are predictive of peat depth. Airborne surveys are used for various applications like mineral prospecting and radon risk management, and some countries have high spatial

coverage of such data (Minasny et al., 2019; Baranwal and Rønning, 2020).

     The effectiveness of DSM depends not only on the methods and data, but also on the characteristics of the mapping area. Norway may be instructive in this respect because its peatlands vary widely across climates and topographies. Norway has 22 peatland mesotopes, with different hydrology, formation, and development – from topogeneous or soligeneous fens to blanket or raised bogs (Joosten and Clarke, 2002; Lyngstad et al., 2023). These mesotopes have fundamentally different

geomorphology, suggesting that peat depth relationships with terrain or radiometric predictors will vary by landscape.

     The little we know about the depths of Norwegian peatlands comes from surveys meant to identify arable land. After scattered surveys in the early 20th century, a comprehensive round of surveying was completed 1964–2001 as part of a wider land cover mapping in Norway (Bjørdal, 2007). This survey produced the initial maps from which Norway's most detailed updated land cover datasets are derived – including the AR5 and DMK datasets used in this study and described later. Because

of its agricultural and silvicultural focus, the survey covered only productive areas below the tree line, and peat depth was measured only in places judged to be potentially arable or afforestable (Ahlstrøm et al., 2019). Field surveyors carried a 1 m probe and they measured peat depth as a categorical variable: shallow (< 1 m), or deep (> 1 m). These classes were assigned



to whole polygons, so spatial resolution is on the order of hectares. The number of measurements per unit area was not standardized but probably low.

The push for nature-based climate solutions motivates for broad mapping of peat depth, to identify peatlands with rich carbon stocks to avoid their conversion and prioritize their preservation or restoration (Strack et al., 2022). Although ground surveys of peat depth are accurate and feasible across small areas, having landscape-scale maps *before* detailed investigation increases the option space for spatial planning and land management. For example, the Norwegian Public Roads Administration routinely measures peat depth during geotechnical work, but by that time, the route of the road is already set. With a digital peat depth

map, planners could better compare routes and climate mitigation measures would have more leverage. Therefore, soil science needs more studies of relationships between peat depth and topographic or radiometric data to determine how predictive they are, which variables are most predictive, and how consistent their associations are.

    Here, we assess how well remotely sensed topographic and radiometric data can predict peat depth at two contrasting sites, with a view towards revising regional and national maps in Norway. Specifically, we aim to: (1) quantify the accuracy of

predictions from topographic and radiometric variables, and (2) identify key predictive variables. Reflecting on our results and the need for continued improvement in peat depth maps, we close by discussing implications for digital soil mapping of peat depth.

    To our knowledge, this is the first study to predict peat depth from airborne radiometric data using machine learning algorithms. Where airborne radiometric data have previously been used to predict peat depth, it has been through modeling

techniques less suited for prediction and spatial extrapolation (e.g., Keaney et al., 2013; Gatis et al., 2019; Siemon et al., 2020). Where machine learning algorithms have previously been used on airborne radiometric data, it has been to predict peat extent rather than depth (e.g., O'Leary et al., 2022).

## 2   Materials and methods

### 2.1   Sites

To assess the predictive value of terrain and radiometric data for mapping peat depth, we chose two sites with conspicuously different physical geography: Skrimfjella (Fig. 1b) in eastern Norway and Ørskogfjellet (Fig. 1a) in western Norway (Fig. 1c). These sites were chosen because they were covered by radiometric data from airborne surveys, had relatively little built-up area, and were accessible by road.

    At Skrimfjella we delineated a study area of 34 km$^2$ based on radiometric coverage and accessibility (Fig. 1b). The study

area has a diverse bedrock, with 32 % alkali feldspar granite, 26 % marlstone, 10 % granite, 8 % monzonite, 7 % sandstone, 6 % limestone, and five other rock types with 1–5 % coverage (Geological Survey of Norway, 1:250 000 dataset). The area within our delineation at Skrimfjella is classified as the landscape type *inland hills and mountains* (Simensen et al., 2021). It is almost without human infrastructure, dominated by conifer forest, and borders on a nature reserve. The study area has a mean elevation of 438 m a.s.l. (range 223–711 m, IQR 351–509 m), and its mean slope at 10 m resolution is 10.8° (IQR 4.6–15.1°).





**Figure 1.** Sites in southern Norway (a) with land cover at Skrimfjella (b) and Ørskogfjellet (c). To fit the scale of the maps, the land cover shown here is from the AR50 national land resource dataset, which has simplified geometry with respect to the AR5 dataset that is used in the study. Terrain is visualized using a hillshade with slope 45° and azimuth 225°. AR50 data are from Norwegian Institute of Bioeconomy Research under the Norwegian Licence for Open Government Data (https://data.norge.no/nlod/en/1.0) and terrain data are from the Norwegian Mapping Authority under the Creative Commons Attribution 4.0 International license (https://creativecommons.org/licenses/by/4.0/).



In Norway's AR5 national land cover dataset ("areal resources in scale 1:5000", Ahlstrøm et al., 2019), 1.5 km$^2$ (4.5 %) of the study area is classified as 'mire' – defined as areas with mire vegetation and at least 30 cm of peat depth.

At Ørskogfjellet we defined a study area of 124 km$^2$ which basically followed the southernmost part of the radiometric survey extent (Fig. 1a). Bedrock in the area is 84 % granitic gneiss, 11 % granite, and 5 % aluminium silicate gneiss (Geological Survey of Norway, 1:250 000 dataset). This study area comprises a wide range of major landscape types: *coastal plains*, *coastal fjord*,

*inland valleys*, as well as *inland hills and mountains* (Simensen et al., 2021). It is mostly forested, but also contains considerable farmland and open upland, and has several large lakes. Its mean elevation is 211 m above sea level (range 0–807 m, IQR 73–310 m), and its mean slope at 10 m resolution is 13.0° (IQR 4.7–18.3°). In the AR5 dataset, 15.3 km$^2$ (12.4 %) of the study area is classified as mire.

## 2.2   Peat depth predictors

We created the same suite of peat depth predictors for both sites (25 continuous and 1 categorical; Table 1). Each continuous predictor equates to one variable in the model, while the categorical predictor equates to two variables in the model because its three levels become two indicator variables (one level is used as the reference). All continuous predictors were derived either from an airborne radiometric survey or from a DTM. From the radiometric surveys, we simply used the four variables produced by the Geological Survey of Norway: ground concentration of potassium, thorium, uranium, as well as total count. From the

DTMs we calculated several land surface parameters, ranging from simple terrain indices to more complex geomorphometric and hydrological variables (Maxwell and Shobe, 2022). The categorical predictor was peat depth class, from a national map dataset. We chose a spatial resolution of 10 m for our predictors, peat depth sample, and modelling. We considered this a reasonable compromise between DTM resolution (1 m) and small peatlands on the one hand, and airborne radiometric resolution (50 m) on the other hand. Complete descriptions of predictors follow below.

### 2.2.1   Radiometric

The Geological Survey of Norway conducted and processed radiometric surveys over our study areas, as reported in Baranwal et al. (2013) and Ofstad (2015). They provided us for each site four variables at 50 m resolution, which we downscaled to 10 m resolution by cubic spline resampling, using the *terra* package (v1.7) in R.

The radiometric surveys were conduced with similar flight parameters (Table A1). Spectrometer count rates were calibrated

to known concentrations of potassium, thorium, and uranium in mobile pads. Data were processed following standard procedures outlined by the International Atomic Energy Association. Processing included: correction for aircraft and cosmic background radiation, correction for radon in the air, window stripping of the gamma ray spectrum, correction for flying height, conversion of count rates to ground concentrations, and finally gridding to 50 m resolution with micro-leveling. At Ørskogfjellet an additional convolution filter was added to smooth the gridded data.



**Table 1.** Twenty-six candidate predictors of peat depth. Note that each continuous predictor contributes one variable to the model while the single categorical predictor (DMK) contributes two binary variables, corresponding to its three levels (one level is used as the reference). This results in a total of 27 variables.

| Group | Code | Units | Description |
|---|---|---|---|
| radiometric | radK | % | Potassium ground concentration |
| | radTh | ppm | Thorium ground concentration |
| | radU | ppm | Uranium ground concentration |
| | radTC | $\text{counts}\,\text{s}^{-1}$ | Total Count of gamma radiation |
| terrain | elevation | m | mean elevation from DTM with 1 m resolution |
| | slope1m | degrees | mean of slope at 1 m resolution |
| | TPI1m | m | mean of Topographic Position Index at 1 m resolution |
| | TRI1m | m | mean of Terrain Ruggedness Index at 1 m resolution |
| | roughness1m | m | mean of roughness at 1 m resolution |
| | slope10m | degrees | slope from DTM with 10 m resolution |
| | TPI10m | m | Topographic Position Index from DTM with 10 m resolution |
| | TRI10m | m | Terrain Ruggedness Index from DTM with 10 m resolution |
| | roughness10m | m | roughness from DTM with 10 m resolution |
| | MRVBF | dimensionless | Multi-Resolution Valley Bottom Flatness |
| | TWI5m | dimensionless | mean of Topographic Wetness Index at 5 m resolution |
| | TWI10m | dimensionless | Topographic Wetness Index at 10 m resolution |
| | TWI20m | dimensionless | bilinear interpolation of Topographic Wetness Index at 20 m resolution |
| | TWI50m | dimensionless | bilinear interpolation of Topographic Wetness Index at 50 m resolution |
| | DTW2500 | m | Depth-To-Water index, flow initiation area of 0.25 ha |
| | DTW5000 | m | Depth-To-Water index, flow initiation area of 0.5 ha |
| | DTW10000 | m | Depth-To-Water index, flow initiation area of 1 ha |
| | DTW20000 | m | Depth-To-Water index, flow initiation area of 2 ha |
| | DTW40000 | m | Depth-To-Water index, flow initiation area of 4 ha |
| | DTW80000 | m | Depth-To-Water index, flow initiation area of 8 ha |
| | DTW160000 | m | Depth-To-Water index, flow initiation area of 16 ha |
| DMK | DMK | categorical | peat depth class in the DMK national map dataset: shallow/deep/unknown |





### 2.2.2 Terrain


For terrain-derived predictors, we obtained 1 m resolution rasters from the national DTM (CC BY 4.0 Norwegian Mapping Authority). The DTM for Skrimfjella was produced from airborne laser scanning surveys in 2015 and 2022, with laser point density of $5\,\mathrm{pts\,m^{-2}}$. For Ørskogfjellet, the DTM was produced from a 2015 survey with $2\,\mathrm{pts\,m^{-2}}$. Where necessary, DTMs were resampled to the coordinate reference system of the radiometric data.

We used the *terra* package to calculate from the DTMs: slope, Topographic Position Index (difference from mean of eight neighbors), Terrain Ruggedness Index (mean of absolute differences from eight neighbors), and roughness (range in the nine-cell neighborhood). These were derived at two scales to produce eight different predictors; we either calculated the indices at 1 m DTM resolution and then aggregated to 10 m resolution, or aggregated to 10 m DTM resolution and then calculated the indices. This kind of multiscale feature engineering of land surface parameters has been found to improve machine learning

predictions of soil properties (Miller et al., 2015; Dornik et al., 2022; Newman et al., 2023). We know that peat depth tends to vary at fine scales in Norway, which is why we chose 1 m and 10 m resolutions (Maxwell and Shobe, 2022). We also calculated the Multi-Resolution Valley Bottom Flatness index, which indicates the degree of valley bottom flatness at a given location via a multiscale algorithm (Gallant and Dowling, 2003). We calculated this index in SAGA GIS (v.9.3.2, Morphometry library, Conrad et al., 2015) with default parameters (initial slope threshold = 16 %, lowness threshold = 0.4, upness threshold = 0.35,

slope shape parameter = 4, elevation shape parameter = 3).

The Topographic Wetness Index (Quinn et al., 1991) is notoriously scale-dependent and often matches real hydrological conditions best when calculated from moderate to coarse resolution DTMs (Ågren et al., 2014; Riihimäki et al., 2021), so we calculated it from 5 m, 10 m, 20 m, and 50 m DTM resolution. The calculations were performed with Whitebox software (Lindsay, 2016a), accessed through the *whitebox* R package (v2.4, Wu and Brown, 2022). We filled depressions in the DTM

with the algorithm in Wang & Liu (2006), and used the deterministic infinity flow accumulation algorithm (Tarboton, 1997).

The Depth-to-Water index (Murphy et al., 2007) approximates a location's vertical height above the surface water feature (e.g. stream, lake, or sea) that it is likely to drain towards. It is calculated as the minimum cumulative slope (scaled by cell size) to a surface water feature (eq. 5 in Murphy et al., 2009). We calculated unitless slope from the 1 m DTM using the Whitebox software. Also using Whitebox, we defined surface water features from the DTM by filling depressions and then

calculating flow accumulation to define catchment areas for each cell (Schönauer et al., 2021; Schönauer and Maack, 2021). This catchment area layer was then thresholded at seven different levels (flow initiation area 0.25–16 ha) to estimate surface water features under moisture scenarios varying from wet to dry (Murphy et al., 2011; Ågren et al., 2014; Schönauer et al., 2021). In addition, all surface water features mapped in the AR5 dataset were also transferred to the raster layer. For each of the seven surface water layers, we derived Depth-to-Water using the *Distance Accumulation* tool in ArcGIS Pro (v.3.1, ESRI,

USA), which has an efficient algorithm to find the cumulative distance over a cost surface to the least-cost source.





### 2.2.3 Peat depth class

We prepared one categorical predictor – peat depth class – from the national map dataset called *DMK* (Ahlstrøm et al., 2019). The DMK dataset is derived from the same historical surveys as the AR5 dataset, and peat depth classes are: < 1 m (*shallow*), > 1 m (*deep*), and *unknown*. Surveyors generally assigned peat depth classes to polygons of at least 0.5 ha, although delineating polygons down to 0.2 ha was allowed if peat depth showed a "particularly marked difference" (Bjørdal, 2007). We rasterized the peat depth class attribute to our 10 m grid.

### 2.3 Peat depth sample selection

The places (10 m raster cells) we chose to measure peat depth were sampled from mire areas in the AR5 dataset, and optimized for training a Random Forest (RF) model of peat depth (Brus, 2019). Broadly, we aimed for a sample that was representative of the predictor space defined by the most important predictors of peat depth (Wadoux et al., 2019; Ma et al., 2020). A sample that preserves the properties of the multivariate distribution of predictor and outcome variables is most likely to maintain any complex, non-linear relationships that exist in the population, while avoiding spurious ones (Brus, 2019). Although we implemented the approach differently for Skrimfjella (in 2020) than for Ørskogfjellet (in 2023), the objective was the same.

We made several point measurements of peat depth within the sampled raster cells (three if by probing only, more if by GPR as well). The point measurements (described below) were ultimately aggregated to 10 m resolution by taking the mean of point values within each cell, inversely weighted by their distances to the cell center.

### 2.3.1 Skrimfjella

At Skrimfjella, we used the *eSample* function in the *iSDM* R package (v1.0) to stratify our sample across elevation, slope, and potassium concentration. This function defines the environmental space as a two-dimensional convex hull around the PCA-ordinated data, then creates a regular grid across that space, and lastly finds for each grid cell the datum that is nearest (Hattab et al., 2017). We set a target sample size of 100, excluded the top and bottom percentile from the convex hull, and *eSample* returned 105 raster cells.

We also measured peatland occurrence (binary variable) in a separate sample at Skrimfjella. We measured peatland occurrence because the AR5 dataset is known to underestimate peatland extent (especially in forests, Bryn et al., 2018), and because airborne radiometrics may help identify unmapped peatland (Gatis et al., 2019; O'Leary et al., 2022). The 10 m cells were sampled from those outside AR5 mire and with slope < 20°. Following the same procedure, *eSample* returned 106 raster cells from this population.

### 2.3.2 Ørskogfjellet

At Ørskogfjellet, we first determined a minimal sample size that would adequately capture the slope and radiometric properties (potassium, thorium, uranium, and total count) of the entire AR5 mire area (Saurette et al., 2023). Specifically, we identified an elbow point in a curve of similarity between sample and population (Malone et al., 2019). For a sequence of sample sizes (50–





500) (ten replicates each, drawn by conditioned latin hypercube sampling, Minasny and McBratney, 2006; Roudier, 2011), we calculated the mean Kullback–Leibler divergence between sample and population distributions (Malone et al., 2019; Saurette et al., 2023). Then we fitted an asymptotic regression of mean divergence on sample size, and found that the curve reached 95 % of the fitted asymptote at a sample size of 160.


To choose 160 locations, we performed feature space coverage sampling, implemented using the *kmeans* function in base R and the *rdist* function in the the *fields* package (v14.2). Feature space coverage sampling chooses locations that are closest to cluster centers in standardized predictor space (Brus, 2019). This approach has been found to produce higher accuracy in RFs than conditioned latin hypercube sampling (Wadoux et al., 2019; Ma et al., 2020). Feature space coverage sampling works best when all dimensions are important predictors of the outcome (Wadoux et al., 2019), and we used the same five predictors that we used to choose sample size: slope and four radiometrics.


We adjusted the feature space coverage sampling to ensure that locations were accessible within time constraints, and assessed how this changed our sample from an ideal feature space coverage sample. Adjusting for accessibility is justified because the smaller sample size that would result if accessibility were ignored can degrade model accuracy as much or more as deviations from ideal sampling designs (Wadoux et al., 2019; Ma et al., 2020). To adjust, we first restricted the sampling population to AR5 mire areas that were within an arbitrary cost distance of publicly accessible roads. Cost distance was calculated using GRASS's *r.walk* function, with friction costs defined by AR5 land classes (GRASS Development Team, 2022). After creating a feature space coverage sample with this restriction, we also inspected a map of the sample and substituted 16 inaccessible locations with accessible locations from the same or a nearby cluster. Our two accessibility adjustments increased the distance in standardized predictor space between sample locations and cluster centers by 78 % (compared to the ideal sample), but distance in our sample was still only 46 % of the mean distance to cluster centers – i.e., accessibility did not force locations far from cluster centers relative to the size of the clusters.



## 2.4   Depth measurements

We measured peat depths at Skrimfjella in August 2020 and at Ørskogfjellet in August 2023. At both sites we used manual probing as the primary method of measuring peat depth, and ground-penetrating radar (GPR) as a secondary method. That is: peat depth was always measured by probing, and also by GPR in a subset of cells that partly overlapped with the probed cells We chose this approach because probing is a fast and reliable method for point measurements, while GPR can provide higher lateral density of data in the same amount of time (Parry et al., 2014). Probed depths serve to calibrate GPR measurements when calibration by common midpoint survey is not possible – as was the case with our fixed radar antennas – so the methods are complementary.



### 2.4.1   Peat probing

We navigated to the centers of the raster cells in our samples using handheld (Skrimfjella) or real time kinematic (Ørskogfjellet) global navigation satellite system (GNSS) receivers. We dampened the effect of outlying measurements by probing three times at each location (Parry et al., 2014), at the vertices of a centered triangle with 2 m (Skrimfjella) or 4.5 m (Ørskogfjellet) sides.





We used changes in resistance to indicate the base of the peat column. Probe locations were adjusted up to 20 cm if the base of the peat column seemed to be blocked by an obvious artifact, like a buried rock. Where the peat column was deeper than the extendable probe could be manually inserted and extracted by a pair of operators, we recorded a right-censored result (one at Skrimfjella, five at Ørskogfjellet).

We also extracted a set of existing depth measurements for Ørskogfjellet from a paper map made by the Norwegian Soil and
Mire Company in 1984. The map presents 44 borehole depths (in decimeters) across a 9 ha peatland area. We georeferenced the map and digitized the borehole locations and depths.

For our sample of peatland occurrence at Skrimfjella, we recorded the presence or absence of peatland at the cell center – primarily by digging and examining the top 20 cm of soil. We judged whether the soil was a peat soil based on its density, texture, and color, as well as the presence or absence of mire vegetation. Although peat soil is strictly defined by organic
content (which we did not analyze), we believe our protocol produced reasonable determinations of presence or absence that would generally satisfy most of the varying definitions of peatland (Minasny et al., 2024a).

### 2.4.2 Ground-penetrating radar

We performed GPR surveys in three subjectively chosen peatlands at Skrimfjella and in areas with a high density of sampled raster cells at Ørskogfjellet. We used the Malå ProEx GPR system (Guideline Geo AB, Sweden) with a GNSS-enabled control
unit connected to a 500 MHz shielded antenna mounted in a plastic sledge (transmitter–receiver separation 0.18 m, trace frequency 10 Hz). For some transects at Ørskogfjellet we substituted a 100 MHz Malå rough terrain antenna (transmitter–receiver separation 2.2 m, trace frequency 5 Hz), because the lower frequency antenna gives greater penetration depth. In all cases the system was towed by a walking GPR operator.

GPR traces were recorded along zigzag (Skrimfjella) or snaking (Ørskogfjellet) transects. At Ørskogfjellet, transects were
predetermined to pass through the centers of sampled raster cells, and we marked these precisely with flags to guide the GPR operator. A GPR records the time taken for a radio wave to travel from the transmitter to a reflector and back to the receiver, and the velocity of the wave varies with properties of the peat column. Therefore, wave velocity has to be calibrated to convert travel time to peat depth, and we probed peat depth along the transects.

We processed the GPR data with Reflex2DQuick (v.3.0; Skrimfjella) or REFLEXW (v.8.5; Ørskogfjellet) software (Sandmeier
Scientific Software, Germany). We applied a time-zero correction, a dewow filter, and a gain filter based on observed energy decay. With Ørskogfjellet data, we also applied a bandpass filter and a dynamic correction that accounts for the non-vertical wave path between offset transmitter and receiver antennas. The latter is important for the rough terrain antenna, where the antenna separation is comparable to typical peat depths. From the processed radargrams, we picked travel times from strong reflectors that we interpreted as the base of the peat column.

We used picks near probed depths to calibrate wave speed velocity – separately for each site. Calibration data were created by matching marked trace locations to a corresponding depth probe (Skrimfjella), or by a spatial join that identified interpreted traces and depth probes within 2 m of each other (Ørskogfjellet). We had sufficient calibration point density to avoid bias in wave velocity as a major source of error (Rosa et al., 2009): 46 calibration points along 3.5 km of interpretable traces at





Skrimfjella, and 78 along 7.8 km at Ørskogfjellet. We fitted site-specific linear regressions of probed depth on picked travel

time, with the intercept fixed at zero, to estimate wave velocities. Notwithstanding a few outlying points, our regressions showed good fits and the resulting velocities are within the range reported for peat (Parsekian et al., 2012): $0.0387\,\mathrm{m\,ns^{-1}}$, $R^2 = 0.874$ at Skrimfjella and $0.0427\,\mathrm{m\,ns^{-1}}$, $R^2 = 0.946$ at Ørskogfjellet. Finally, we used these two wave velocities to convert the travel times of all picks to calibrated peat depths. In total, the GPR surveys produced 48579 point measurements of peat depth at Skrimfjella and 32653 at Ørskogfjellet.

We also used a set of existing GPR depth measurements from Ørskogfjellet, commissioned and provided to us by the Norwegian Public Roads Administration. These data were collected in 2020 and 2021 with a dual channel system (70 MHz and 300 MHz; ImpulseRadar AB, Sweden), connected to GNSS with CPOS correction. For the work presented here, we used a total of 403440 interpreted and calibrated traces along 7.4 km of transects – discarding data where multiple depths were interpreted for the same location.

## 2.5 Predictive models of peat depth

### 2.5.1 Modelling approach

We used Random Forests (RF) to predict peat depth at both sites. RF is a tree-based ensemble machine learning algorithm that builds many decision trees on bootstrapped samples of the training data, randomly subsets predictors in the trees, and averages the predictions of the trees (Breiman, 2001). We chose RF because it can handle complex interactions between predictors, is

robust to overfitting, and generally shows higher performance in DSM applications than other algorithms (Beguin et al., 2017; Nussbaum et al., 2018; Lamichhane et al., 2019). It is suited for use on relatively small training datasets and its predictions can be interrogated to learn about predictor importance (Khaledian and Miller, 2020). Evaluating variable importance in a maximally predictive model aligns with the aim of this study.

RF by itself is not a spatial model, and it will only predict spatial structure in the outcome to the degree that the structure

is captured by predictors. We considered using regression kriging – a hybrid between non-spatial and spatial techniques that would be achieved by adding to the RF predictions a geostatistically interpolated surface of RF residuals (Hengl et al., 2004). The spatial component in regression kriging often improves map accuracy compared to a non-spatial model (Beguin et al., 2017; Lamichhane et al., 2019; Molla et al., 2023), but it can do so only if the spatial autocorrelation range in the non-spatial residuals is large compared to distances between samples and prediction locations (Hengl et al., 2004; Szabó et al., 2019;

Takoutsing and Heuvelink, 2022). If the outcome varies at fine scales and the samples are clustered in small parts of the study area, a spatial component will hardly improve overall map accuracy. We used semivariograms to assess the spatial structure in the residuals of the RF predictions, and found that (non-spatial) RF rather than regression kriging was justified at both sites.

We implemented models in the *tidymodels* framework in R (Kuhn and Wickham, 2020), with the *ranger* R package for RFs (v0.16, Wright and Ziegler, 2017). RFs were fit with 1000 trees, minimum node size of 5, and the number of predictors

randomly sampled at each split was the square root of the total number of predictors (*ranger* default). We did not tune these





hyperparameters because RFs are relatively insensitive to tuning (Probst et al., 2019), and because it would require nested spatial cross-validation to prevent data leakage (Schratz et al., 2019).

### 2.5.2 Model performance

For both sites, we compared the performance of models with five different configurations of predictors. Specifically, we trained
models with:

1. *DMK* (2 variables)
2. *terrain* (21 variables)
3. *terrain + DMK* (23 variables)
4. *terrain + radiometric* (25 variables)
5. *all predictors* (27 variables)

These different configurations simulate different scenarios of data availability. Comparing the different configurations allowed us to isolate the added value of each of the predictor groups. We did not explore configurations comprising radiometrics without terrain, because LiDAR terrain surveys typically precede airborne radiometric surveys. The models with only DMK peat depth class were simple linear models rather than RFs, and served to provide a fair comparison between the accuracy of the RF
models and the existing national map of peat depth, calibrated on the same data.

We used a spatial cross-validation scheme to evaluate model performance (Wadoux et al., 2021; Meyer and Pebesma, 2022). To set the folds we used k-Means Nearest Neighbor Distance Matching (kNNDM), which aims to mimic the spatial prediction task that is defined as the goal (Linnenbrink et al., 2024). In particular, kNNDM looks for the spatial assignment of training data to folds that minimizes the difference between two distributions: nearest neighbor distances between training and test locations
in the cross-validation, and nearest neighbor distances between training and prediction locations for the model. That way, the spatial separation between folds is similar to the separation between training and prediction locations – which increases the quality of the map accuracy estimate (Linnenbrink et al., 2024). For spatially clustered training data, this approach strikes a balance between the risk of optimistic metrics from random cross-validation and the risk of pessimistic metrics from other forms of spatial cross-validation (Wadoux et al., 2021). We implemented the kNNDM with the *CAST* R package (v1.0, Meyer
et al., 2024), setting prediction locations to all AR5 mire cells in the study area, and choosing a number of folds (5–20) that produced the best match between the two NND distributions. From the cross-validation we quantified mean absolute error (error magnitude, original scale), $R^2$ (explained variation, standardized scale), and Lin's concordance correlation coefficient (error magnitude and explained variation, standardized scale).

DSM products have much more value when their predictions are accompanied by uncertainty estimates, and all DSM should
strive to assess uncertainty (Arrouays et al., 2020; Wadoux et al., 2020) and evaluate uncertainty estimates (Heuvelink and Webster, 2022). Therefore, we produced prediction intervals with quantile regression forests (Meinshausen, 2006), and used the same spatial cross-validation to evaluate the prediction interval coverage probability (Shrestha and Solomatine, 2006). The



quantile regression forests were trained with predictor configuration that showed the highest performance at each site (under the assumption that these models would be put into production) and we extracted 90 % prediction intervals.

### 2.5.3 Peatland extent

While the primary aim of the DSM was to predict peat depth within AR5 mires (where peat depth is supposed to be > 30 cm), we also tested whether our mire-trained models of peat depth could identify peatlands outside of AR5 mires. For both sites we performed an evaluation with our (cell-aggregated) peat depth measurements, of which about one-fifth were outside of AR5 mires. For this evaluation we used the same spatial cross-validation folds as previously, but now we trained the models only on AR5 mire locations (in the training folds), and evaluated them only on (test fold) locations classified in AR5 as something other than mire. From the cross-validation we quantified mean absolute error. For Skrimfjella – where we measured peatland occurrence separately from depth – we also trained a model on all peat depth measurements (AR5 mire or not) and then used the independent occurrence dataset to evaluate this model. Specifically, we calculated the area under the curve of the receiver operating characteristic, to evaluate the model's ability to discriminate between peat presence and absence. This analysis treats the model's prediction of peat depth as a rank index of peatland likelihood. Since the purpose was to test the models' ability to uncover unmapped peatland (where DMK peat depth class is undefined), we evaluated models with two predictor configurations: *terrain* or *terrain + radiometric*.

### 2.5.4 Model interpretation

We quantified global variable importance (predictor influence across all locations) and examined partial dependence plots (curves of fitted relationships) for the best-performing predictor configuration at each site. Both are useful for understanding the mechanisms behind the model's predictions and the roles of the predictors in the model.

For both sites, we interpreted a model trained on a non-collinear subset of variables from the best performing predictor configuration – because correlation between predictors degrades variable importance metrics (Strobl et al., 2008; Biau and Scornet, 2016) and can produce misleading visualizations of predictor–outcome relationships (Biecek and Burzykowski, 2021; Dwivedi et al., 2023). Specifically, we eliminated variables from the best performing predictor configuration to obtain a set with no pairwise Pearson correlation coefficient above 0.7. Thus, highly correlated sets of variables are represented by a single variable for the purposes of model interpretation.

We calculated variable importance with the *vip* R package (v0.4), by three different methods: *FIRM*, *permutation*, and *Shapley* (Greenwell and Boehmke, 2020). *FIRM* values measure the flatness of the partial dependence plot, *permutation* values measure the decrease in model performance when the predictor is permuted, and *Shapley* values are aggregated from local, game-theoretical measures of variable importance (Greenwell and Boehmke, 2020). *Permutation* values were obtained from ten iterations, with root mean square error as the performance measure.

We calculated partial dependence with the *pdp* R package (v0.8, Greenwell, 2017). For the six most important variables, we plotted both partial dependence and individual conditional expectation, to show the average effect of the predictor on



**Table 2.** Mean peat depth (cm) in 10 m cells at Skrimfjella and Ørskogfjellet. The cells are also shown stratified by AR5 land class and DMK peat depth class.

|  |  | Skrimfjella | | | Ørskogfjellet | | |
|---|---|---|---|---|---|---|---|
|  |  | n | % | depth | n | % | depth |
| all measured cells |  | 372 | 100 | 119 | 1878 | 100 | 126 |
| AR5 | Agricultural |  |  |  | 21 | 1 | 30 |
|  | Forest | 52 | 14 | 65 | 272 | 15 | 71 |
|  | Open upland | 19 | 5 | 98 | 134 | 7 | 39 |
|  | Peatland | 301 | 81 | 130 | 1451 | 77 | 145 |
| DMK | deep (>100 cm) | 94 | 25 | 100 | 659 | 35 | 219 |
|  | shallow (<100 cm) | 6 | 2 | 50 | 838 | 45 | 82 |
|  | unknown | 272 | 73 | 127 | 381 | 20 | 60 |

the outcome and the variation in the effect across observations, respectively (Goldstein et al., 2015). Non-parallel individual conditional expectation lines indicate the presence of interactions between predictors.

## 3   Results

Our point measurements of peat depth (all sources) produced aggregated depths for 372 cells at Skrimfjella (2.4 % of AR5 mire area) and 1878 cells at Ørskogfjellet (1.2 % of AR5 mire area). Roughly 80 % of these 10 m cells were within AR5 mires, and 385 the remainder in forest, open upland, or farmland (Table 2). Coverage of DMK peat depth classes was higher at Ørskogfjellet than at Skrimfjella (79 % versus 27 % of cells), and at Ørskogfjellet the *deep* and *shallow* classes showed a larger difference in measured depth. Overall mean peat depths were similar at Skrimfjella and Ørskogfjellet: 119 cm and 126 cm, respectively.

### 3.1   Model performance

None of the models were able to predict peat depth across the study areas with high accuracy (Fig. 2). For Skrimfjella, the 390 best model achieved a concordance correlation coefficient of 0.30, an $R^2$ of 0.34, and a mean absolute error of 60 cm. For Ørskogfjellet, the concordance correlation coefficient was 0.39, $R^2$ was 0.33, and mean absolute error was 56 cm, so the best model at Ørskogfjellet was slightly more accurate than the best model at Skrimfjella. These values were derived from kNNDM spatial cross-validation with 20 folds at Skrimfjella and 10 folds at Ørskogfjellet.

Although the difficulty of predicting peat depth caused prediction intervals to be wide, these uncertainty estimates were 395 well calibrated. At Skrimfjella, the prediction interval coverage probability was 91 %, and at Ørskogfjellet it was 84 % (both compared to the target value of 90 %). Observations outside of the prediction intervals were evenly distributed across each of the study areas.





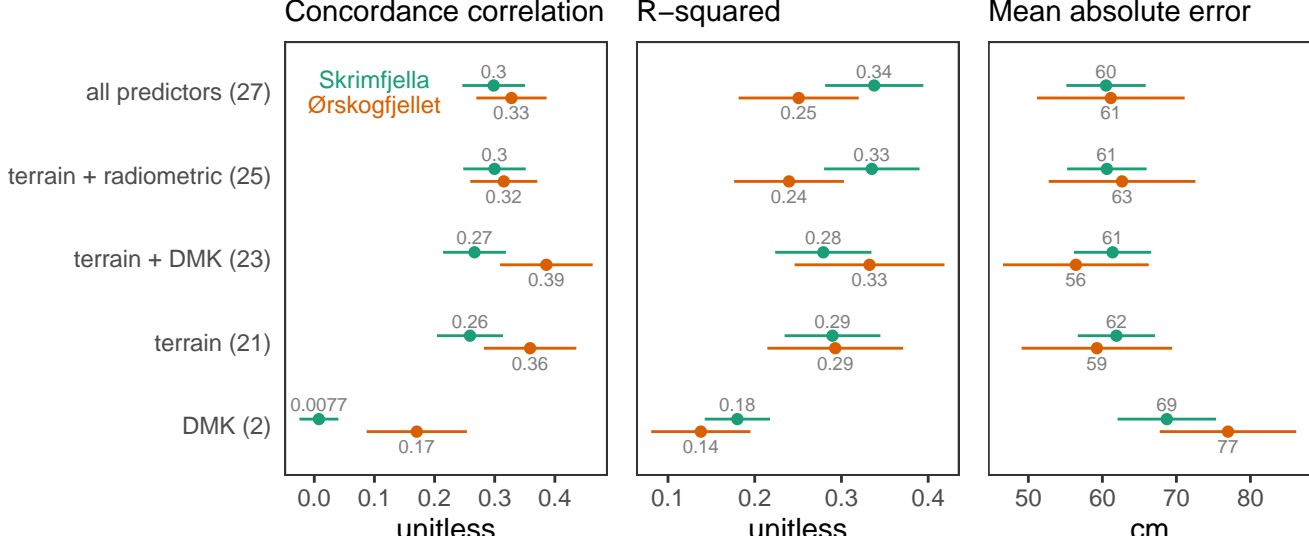

**Figure 2.** Performance of peatland depth models with different predictor configurations, evaluated via spatial cross-validation. Parentheses denote the number of variables in each predictor configuration, and point estimates are shown +/- their standard error.

For Skrimfjella, the best predictor configuration was *all predictors*, followed closely by *terrain + radiometric*. The performance gap between the *terrain + DMK* configuration and the *terrain* configuration was similarly small. Compared to the *terrain*

configuration, the *terrain + radiometric* configuration improved concordance correlation by 0.04, $R^2$ by 0.04, and mean absolute error by 1 cm. *DMK* was a very poor predictor of peat depth even though it was calibrated to measured depths, with a concordance correlation coefficient of 0.0077.

For Ørskogfjellet, the best predictor configuration was *terrain + DMK*, followed by *terrain*. Adding radiometric predictors to these configurations worsened model performance, especially in terms of concordance correlation and $R^2$. By itself, *DMK*

*class* produced a concordance correlation coefficient of 0.17 (compared to 0.008 at Skrimfjella), but the worst mean absolute error of any model at either site: 77 cm.

The best models at both sites overpredicted shallow peats and strongly underpredicted very deep peats (Fig. 3). The mean error (bias) of these models was 10 cm at Skrimfjella and -4 cm at Ørskogfjellet.

### 3.1.1    Peatland extent

When we tested how well models extrapolated from within to outside of AR5 mire areas, we found that the models produced worse mean absolute error than just assuming a constant 30 cm depth (Fig. 4). With the independent occurrence data at Skrimfjella, we found that neither the *terrain* nor *terrain + radiometric* configurations were able to discriminate between





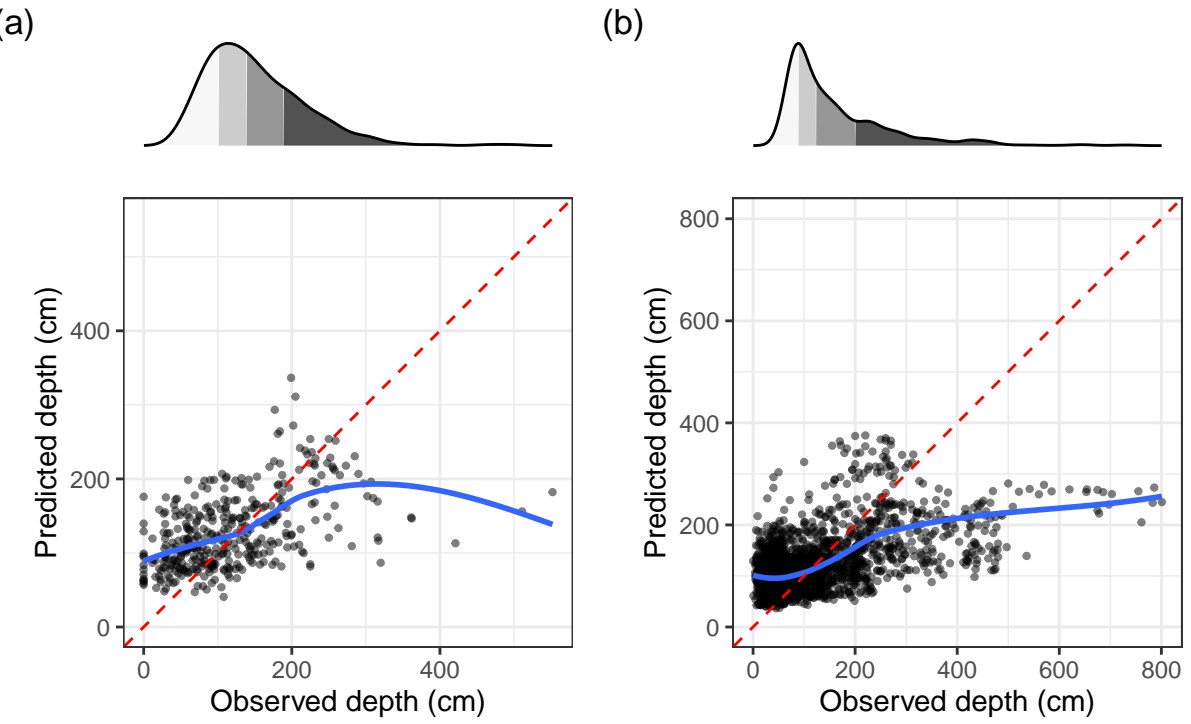

**Figure 3.** Calibration plots for the best-performing models at Skrimfjella (a) and Ørskogfjellet (b), with predictions from spatial cross-validation. Points are transparent to show overlap. Blue lines are local polynomial regressions and the red dashed line in each panel shows the 1:1 line. Marginal distributions (top) are shaded by quartile.

peat presence and absence (area under the curve of the receiver operating characteristic 0.44 and 0.52 respectively, where 0.5 indicates random guessing).

### 3.2 Model interpretation

For the purpose of model interpretation, the *all predictors* configuration for Skrimfjella was reduced from 27 variables to 11 non-collinear variables, by removing one of the variables in each highly-correlated pair. Similarly, the *terrain + DMK* configuration for Ørskogfjellet was reduced from 23 variables to 11 non-collinear variables.

#### 3.2.1 Variable importance

At both sites, elevation and Multi-Resolution Valley Bottom Flatness were important predictors (Fig. 5). At Skrimfjella these two predictors were of similar importance, while at Ørskogfjellet elevation was more important than Multi-Resolution Valley Bottom Flatness. DMK was also important – the shallow class in particular – but only at Ørskogfjellet. Some realizations of the hydrological predictors Topographic Wetness Index and Depth-to-Water showed considerable importance, while others showed





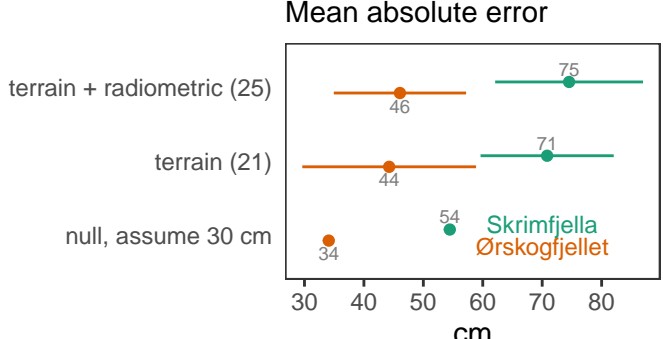

**Figure 4.** Performance of models that extrapolate from training data inside of mapped mires to test data outside of mapped mires (AR5 dataset), evaluated via spatial cross-validation. Parentheses denote the number of variables in each model, and point estimates are shown +/-their standard error.

little – with no consistency between sites. For example, TWI5m, DTW40000, and DTW2500 rounded out the top five most

important variables at Skrimfjella, while TWI20m and TWI50m were least important, other than DMK. At both sites, the most important realizations of hydrological predictors were more important than the simple terrain indices slope, Terrain Ruggedness Index, Topographic Position Index, and roughness. The radiometric predictor uranium ground concentration showed moderate importance at Skrimfjella.

### 3.2.2 Partial dependence

Many of the most important predictors in the best performing models showed non-monotonic effects on peat depth (Fig. 6). At Ørskogfjellet for example, increasing elevation was predictive of deeper peat up to about 75 m above sea level, after which a further increase was predictive of shallower peat. At Skrimfjella the partial dependence on elevation had the opposite shape but covers a higher elevation range, with the shallowest peats predicted around 350 m above sea level. TWI50m at Ørskogfjellet and DTW4000 and uranium ground concentration at Skrimfjella were other predictors that showed considerable fluctuations in

their predictive effects on peat depth. The radiometric predictor in particular displayed an idiosyncratic effect, with a marked dip in predicted depth at intermediate values of uranium ground concentration. On the other hand, the partial effects of some important predictors were more straightforward. The partial dependence on Multi-Resolution Valley Bottom Flatness was quite similar across sites, with the deepest peats predicted in the very flattest valley bottoms. Also, TWI5m and DTW2500 at Skrimfjella showed monotonically positive and negative predictive effects, respectively.

Individual conditional expectation lines indicated some interactions between predictors (Fig. 6). For example, the magnitude of the increase in depth with elevation that the model expected at Ørskogfjellet was different for different observations; some depth predictions increased by only 40 cm while others increased by more than 100 cm, over the same elevation gain. Similarly, individual conditional expectation lines of uranium ground concentration at Skrimfjella were non-parallel, with some locations showing monotonically increasing peat depth predictions with uranium concentration (unlike the average effect). Nevertheless,





**Figure 5.** Global variable importance in the best-performing models at Skrimfjella (a) and Ørskogfjellet (b), as measured by three different metrics: Shapley values, permutation importance, and the Feature Importance Ranking Measure (Greenwell and Boehmke, 2020). Variables removed due to collinearity are shown to the right of the variable with which they are most correlated.





most individual conditional expectation lines were generally parallel – indicating that the average effects of the predictors were
good representations of their overall effects.

## 4 Discussion

### 4.1 Can we improve Norway's peat depth maps?

#### 4.1.1 Remotely sensed variables are weak (but not useless!) predictors of peat depth

Our ability to predict peat depth in the study sites based on terrain and radiometric data was limited. Mean absolute errors of
60 and 56 cm at the two sites — relative to mean depths of 119 and 126 cm — illustrate the practical limitations of these maps.
Since any given 10 m cell will miss by about 60 cm, applications requiring detailed peat depth in a small area (e.g. < 1 ha)
would benefit from measuring depth on the ground rather than relying on the DSM alone.

Ground-based measurements also have uncertainty (Parry et al., 2014), but our data show that this error was smaller than the
variation in peat depth over short distances (e.g. 100 m). Independent probing and GPR measurements up to 2 m apart were
differed by an average of 29–42 cm (Fig. A1). By comparison, the average variation among measurements 100 m apart was
70–106 cm (Fig. B1). We also note that cell-level measurement error that is correlated with predictors (e.g. underestimated
depth at high Multi-Resolution Valley Bottom Flatness), will go undetected by cross-validation and does not account for the
limited accuracy of our models. Ultimately, the effect of measurement error falls mostly outside the scope of DSM, since it can
only be evaluated with higher-quality, independent data.

Although model performance was limited, we improved on the best available map of peat depth (DMK depth class), which
is based on field measurements only. This highlights the general value of remotely sensed data, whose complete coverage can
improve maps even when their association with the variable of interest is weak (Mulder et al., 2011). Since remotely sensed
data are widely available, improvements to soil maps as shown here are low hanging fruit. This point is recognized and reflected
in the rise of DSM (Minasny et al., 2019).

DMK peat depth classes were a worse predictor of peat depth than our models even though we calibrated them with the
same data (Fig. 2). That is: we gave DMK and the DSM equal footing for a fair comparison. If we had taken the DMK peat
depth classes at face value and assumed depths according to their class definitions (< 1 m, > 1 m), they would have performed
worse and the advantage of the DSM would be greater. The advantage of the DSM was not large in absolute terms (9 cm and
21 cm improvements in mean absolute error), but it explained much more of the variation in depth (improvements in $R^2$ of
0.16 and 0.19). We attribute this result to the poor spatial and thematic resolution of DMK peat depth, which precludes a robust
correlation with peat depths varying from 0–8 m at fine spatial scales.

The performance of our terrain- and radiometric-based maps could have been improved with a spatially explicit mapping
approach like regression kriging. By ignoring spatial autocorrelation in peat depth (i.e. the n in *scorpan*), we have not extracted
all of the information about peat depth in the study area out of the training data. If we were to put our DSM approach into
production for published maps, we would harness the spatial component, but the actual maps for Ørskogfjellet and Skrimfjella





**Figure 6.** Partial dependence plots of the six most important variables in the best-performing models at Skrimfjella (a) and Ørskogfjellet (b). The average effect of the predictor on the outcome (red line) overlays the variation in the effect across observations (grey lines). The variables' training data distributions are indicated with rug plots along the x-axes.





are not of primary interest in this study. Moreover, residuals of the RF predictions showed weak spatial structure at Skrimfjella, and only up to a range of 150 m at Ørskogfjellet. Therefore, the improvement from regression kriging would be small overall and limited to parts of the maps close to measurements.

### 4.1.2 Similar error but less explanatory power in Norwegian peatlands

Compared to other studies using terrain and radiometric data to predict peat depth, our models explained less variability in peat depth but generally produced better or comparable error magnitude ($R^2$ vs. mean absolute error or root mean square error; Wadoux et al. (2022)). It is important to keep in mind that differences in peat depth distributions, spatial scales, and evaluation methods make direct performance comparisons precarious. More standardized reporting would help but not eliminate this 485 consideration. For example, $R^2$ is sensitive to high leverage, extreme values, so it will evaluate a right-skewed distribution differently than a symmetrical distribution. We evaluated our models with respect to the explicit purpose of creating peat depth maps across the study areas, but not all studies tailored evaluation to match an explicitly formulated problem (Milà et al., 2022).

Gatis et al. (2019) used similar predictors and the same spatial grain, finding a much stronger relationship between predicted and observed peat depth ($R^2 = 0.68$). Although their random evaluation data partition could make performance estimates too 490 optimistic (Roberts et al., 2017; Wadoux et al., 2021), the confounding effect of spatial structure is probably small because they used linear regression and few predictors. Their model had limited opportunity to overfit to the spatial structure in peat depth. The most salient difference in Gatis et al. (2019) compared to our study is the character of the study area. They study a flatter area with a higher proportion of peatland cover, and their peatland is primarily blanket bog. The peat surface of a blanket bog is tied more closely to its underlying topography than the peat surface of raised bogs or fens (Lindsay, 2016b), which may 495 make depths in blanket bogs easier to predict. The predominance of fens and smaller peatland extent may have contributed to worse performance in our study.

Marchant (2021) examined a subset of the area studied in Gatis et al. (2019) at 100 m resolution, using splines to relax linearity between radiometry/terrain and peat depth. He found that potassium ground concentration alone predicted peat depth with much higher concordance than our models (concordance correlation coefficient = 0.76) and that elevation alone produced 500 comparable performance (concordance correlation coefficient = 0.27). The root mean square error from these univariate models was 46–68 cm (cf. 78 and 75 cm for Skrimfjella and Ørskogfjellet).

Koganti et al. (2023) had a peat depth distribution and predictors similar to ours, but at much smaller spatial grain and extent. Their training and validation points are closer than the range of spatial autocorrelation in peat depth, so our results are best compared to their non-spatial models. They accounted for more variability in peat depth (adjusted $R^2 = 0.71$) but had larger 505 errors (RSME = 110 cm). Koganti et al.'s (2023) linear regression models produced negative predictions, and it is unclear whether the values quoted above include these. If we disregard the negative predictions, their model showed the same pattern as ours in overpredicting shallow peats and underpredicting deep peats, although their underprediction was less severe. An important difference between Koganti et al. (2023) and our study (besides spatial scale) is that they measured radiometrics on the ground, rather than using airborne survey data. Thus, the footprint of their detector was much smaller and they could 510 capture variation in radiation at finer scales.



### 4.1.3 Asymmetries in depth predictions for land use planning and carbon accounting

Our models erred most for the deepest peats (Fig. 3). Where overprediction occurred, the error was smaller. This is not unexpected for the right-skewed distributions of peat depth, but it has management implications. Identifying the deepest peats will require additional field work in candidate areas, which could be defined by an upper quantile of predicted depth. Map users should not trust the maps to identify all large carbon stocks, but they can trust that identified large stocks really are large. That makes the map more suited for "red-lighting" than "green-lighting" peatland conversion, for example. Where it does not prohibit conversion (i.e. predicts shallow peat), a ground survey should be done before conversion is allowed (assuming depth is the only consideration). Although this recommendation aligns with the precautionary principle, here we make it on technical grounds based on the maps' characteristics.

### 4.1.4 Depth predictions do not necessarily extrapolate to peat extent

That we measured more than 30 cm of peat in areas not mapped as mire was expected, because AR5 underrepresents peatland extent by about one third (Bryn et al., 2018). For that reason, maps of peatland extent also need revising. Unfortunately, our models did not predict peat occurrence outside AR5 mires. In these areas, assuming a constant 30 cm of peat (the depth threshold used in AR5) produced less error than our models (Fig. 4). Moreover, the performance gap would likely have been larger had our evaluation data been representative of all areas not classified as mire in AR5. Instead they were concentrated near AR5 mires. Most importantly, the independent occurrence data at Skrimfjella showed the model completely failing to predict occurrence, and these data were suited for testing this ability. Although the Ørskogfjellet model may have done better than the Skrimfjella model on a similar independent evaluation set (since it predicted depth better), the improvement would probably be small.

Why wasn't peat occurrence predicted well? The models were trained on fundamentally different populations of locations (~80 % AR5 mire in training vs. 0 % in prediction), so it is not surprising that the associations they learned did not transfer well. Moreover, the ~20 % of training data from outside AR5 mires were incidentally collected (near AR5 mires) and not representative of other land cover classes. In short, the models were blind to the fact that most of both study areas have no peat.

Peatland extent is probably best mapped using different remotely sensed data than we used here (Bakkestuen et al., 2023), and this study's purpose was not to map extent. Nevertheless — as long as the relevant peatland definition includes a depth component — depth predictions (or predictors thereof) should help delineate peatland extent (O'Leary et al., 2022; Beamish and White, 2024). We return to this point in our discussion of implications for digital soil mapping.

## 4.2 Which variables predict peat depth?

### 4.2.1 Airborne radiometrics do not predict Norwegian peat depth

Radiometric data had no predictive value at Ørskogfjellet, while at Skrimfjella they had minor influence in a relatively weak model. The bedrock is more homogeneous at Ørskogfjellet than at Skrimfjella, so uneven radiogenesis is not a viable



explanation for the differences between sites nor the poor performance in general (Beamish, 2014; Reinhardt and Herrmann, 2019). To the degree that radiometrics had predictive value at Skrimfjella, it appears that they were most valuable near the extremes of the depth distribution, since their inclusion improved $R^2$ more than mean absolute error. All four variables were
highly correlated within the peatland parts of our study sites, so there could be no large differences in their predictive value. This contrasts with Koganti et al. (2023), who found that radiometric total count was a much better predictor than potassium ground concentration.

    We suspect that the primary reason for the poor predictive value of the radiometric data was the large footprint of the detector in the airborne survey. With an average flight altitude of 75 m, less than half of the radiation reaching the detector comes from
inside the 100 m diameter circle directly below it (Beamish, 2016; Beamish and White, 2024). The rest of the measured activity integrates a much wider area. For comparison, empirical variograms of peat depth at Skrimfjella and Ørskogfjellet showed no spatial autocorrelation beyond 50 and 75 m (among GPR data) or 110 and 230 m (among 10 m cells). Basically, the airborne radiometric data will not capture large variation over short (< 100 m) distances; the instrument's field of view has a large smoothing effect on the data (Beamish, 2016; Reinhardt and Herrmann, 2019). Different landforms and the changes they
cause in the geometry between the radioactive source and the detector can also distort airborne measurements (Reinhardt and Herrmann, 2019).

    Studies comparing airborne and ground radiometric surveys confirm that they are poorly correlated in low-activity areas like peatlands (Kock and Samuelsson, 2011; Karjalainen et al., 2025). Karjalainen et al. (2025) found that ground-based measurements predicted peat depth better than airborne measurements. Nevertheless, a large radiometric footprint did not
prevent strong associations with peat depth in Gatis et al. (2019) and Marchant (2021), perhaps because the extensive blanket bog landscape in these studies has more gradual changes in depth (Lindsay, 1995). We are unsure whether short-range depth changes explain the weak associations that Siemon et al. (2020) found in a large raised bog.

    Weather conditions varied during the Ørskogfjellet radiometric survey and affected its data (Ofstad, 2015). Thus, uneven snow cover and air moisture may also have masked the soil signal in these data.
We do not believe that the poor predictive value of the radiometric data in this study was caused by fully attenuated radioactivity — at least not in large part. The RF algorithm's flexibility means that radiometrics could be used for shallower peats if they provided predictive value for that part of the depth distribution, but there is no evidence of that in our results. In the partial dependence plot of uranium ground concentration at Skrimfjella, the expected negative relationship between depth and uranium ground concentration cannot be found by ignoring the left (highly attenuated) side of the distribution. About a
quarter of the peats in our study were less than a meter deep, and full attenuation is unlikely for these (Beamish, 2013).

    Although we do not believe full attenuation is the primary reason for poor performance in our study, it may limit peat depth mapping under other circumstances. First-principle calculations suggest that radiation should be 90 % attenuated after about 50–60 cm of typical, wet peat or 85 cm of unnaturally dry peat (Beamish, 2013; Beamish and White, 2024), and some field tests support these values (Billen et al., 2015). It is remarkable that particular studies detected radiation differences up to several
meters deep (Gatis et al., 2019; Koganti et al., 2023), but these may be the exceptions rather than the rule. Perhaps relatively deeper water tables in these study sites (blanket bog, drained fen, Price et al., 2016) contributed to better penetration.



### 4.2.2 Terrain-based variables can predict peat depth

At both our sites, LiDAR-derived terrain variables predicted peat depth much better than radiometric variables (Fig. 5). Elevation was the most important predictor at both sites, and peat depth showed non-monotonic responses to changes in elevation (Fig. 6). We believe that the idiosyncratic elevational relationships we detected are mostly not generalizable beyond the study areas, because we see no simple mechanism (e.g. via climate) to explain the observed patterns. For example, the increase in peat depth from 350 to 700 m.a.s.l. at Skrimfjella is opposite to the general pattern of deeper peats in lowland than upland Norway (Lyngstad et al., 2023). Moreover, elevation at Ørskogfjellet seems to interact with other variables (non-parallel individual conditional expectation lines), complicating its interpretation. Relationships between elevation and peat depth have previously shown opposite shapes in different areas (Finlayson et al., 2021). Nonetheless, a relationship that is not generalizable beyond the mapping area is still useful for DSM, as long as it is evaluated to demonstrate its robustness for the predictive task (e.g. through kNNDM spatial cross-validation).

One interesting feature of the elevational relationships we found may be generalizable: a steep increase in peat depth near the marine limit after the last ice age. At Ørskogfjellet, the marine limit is about 75 meters above today's sea level (Geological Survey of Norway, Høgaas et al., 2012), where the partial dependence plot of elevation shows a sharp increase in peat depth. In areas under the marine limit there has been less time for peat accumulation since the ice sheets retreated, and it is plausible that this makes peats there shallower, all else being equal. This hypothesis is supported by similar findings at a coarser scale in the Hudson Bay Lowlands, where a strong positive relationship between peat depth and distance from the coast can be explained by isostatic uplift and time since peat initiation (Li et al., 2025). We cannot evaluate this effect at Skrimfjella, where the marine limit is below our study area (at 175 m.a.s.l.).

Another influential terrain-based predictor was Multi-Resolution Valley Bottom Flatness (Fig. 5). Unlike elevation, it showed a monotonic effect on peat depth: greater valley bottom flatness was always associated with increases in peat depth (Fig. 6). Delineating a valley bottom involves ambiguity, but the Multi-Resolution Valley Bottom Flatness index is a pragmatic approach that considers a location a valley bottom if it is sufficiently low and flat at a particular scale (Gallant and Dowling, 2003). The multiscale nature of the index allows small elevated but flat areas (including saddles) to be characterized as having high valley bottom flatness (Gallant and Dowling, 2003). Our results suggest that Multi-Resolution Valley Bottom Flatness is a robust indicator of high water tables (and peat accumulation) over millennial time scales, corroborating other studies (Rudiyanto et al., 2018; Deragon et al., 2023).

Other terrain-derived predictors with predictive value in our study are hydrological (Topographic Wetness Index and Depth-to-Water). Notably, slope was inferior to (Ørskogfjellet) or highly correlated with (Skrimfjella) these hydrological indices. Mappers of peat depth should not assume that slope is the best predictor in its class, despite its prevalence in the literature. Wetter locations (high Topographic Wetness Index and low Depth-to-Water) were generally associated with deeper peat, but these relationships were not as strong or consistent as with Multi-Resolution Valley Bottom Flatness (Figs. 5, 6). The optimal scale for Topographic Wetness Index and Depth-to-Water varied, and likely depends on both the dominant peat formation




processes and the typical size of peatland features in a landscape. Including multiple scales of these variables allows the model to capture different hydrological mechanisms operating at different spatial scales.

### 4.2.3 Legacy depth maps have inconsistent predictive value

DMK peat depth class proved an inconsistent predictor of peat depth. At Skrimfjella, it barely improved model performance (Fig. 2). At Ørskogfjellet, it increased performance more, and both indicator variables were among the most important in the 615 model (Fig. 5). We suspect the discrepancy between sites is due to different levels of effort and coverage during the historical surveys; more lowland peatland near agriculture at Ørskogfjellet may have caused more purposeful surveying. This is evidenced by the fact that 73 % of the cells measured at Skrimfjella had unknown depth in DMK, compared to 20 % at Ørskogfjellet. Our results show that classification as deep peat at Skrimfjella had no predictive value – these cells were no different than those classified as shallow or unknown, all else being equal. Interactions between DMK and other variables also underline the 620 inconsistency of DMK depth maps, even within a site. For example, that a peat at Ørskogfjellet was classified as shallow rather than deep did not always cause the model to predict shallower peat – sometimes it increased the predicted depth.

### 4.3 Implications for digital soil mapping of peat depth

The performance gap between the best models and the DMK models shows that peat depth in Norway should be mapped digitally. Anywhere we have some calibrating measurements, we can get better maps than DMK peat depth classes, at 625 low cost. Moreover, DSM methodology can align map products with open science principles by making their production transparent, reproducible, and updatable. The large difference we found between the coverage and quality of DMK peat depth at Skrimfjella versus Ørskogfjellet underlines these advantages. With DSM we can apply the same approach across different areas and make maps with full spatial coverage, continuous values, and validated uncertainty. The rest of this section discusses recommendations and needs for more extensive peat depth DSM.

### 4.3.1 Peat depth measurements should be organized

High quality DTMs are available for mainland Norway, making peat depth measurements the critical training data need. For a given area, some minimum number of measurements is necessary to create meaningful improvement over DMK peat depth. The proportion of peatlands sampled at Skrimfjella was twice that at Ørskogfjellet, but the Skrimfjella model performed worse, which shows that the size of the depth dataset is not all-important. We had the luxury of stratifying our measurements over 635 candidate predictors, and performance may suffer where locations are opportunistic or tied to a sampling design with a different purpose. The magnitude of this penalty will depend on the dataset characteristics, but having enough depth measurements is probably more consequential (Wadoux et al., 2019).

Depth measurements are foundational, and better infrastructure to make these data findable, accessible, interoperable, and reusable would help DSM and other applications. Geoportal access and data exchange standards (like Natural England's for 640 peat surveys, 2023) are important. Peat data often fall through the cracks between geology-oriented and ecology-oriented





archives, but increased awareness of peatland importance is a good impetus to remedy this situation. Peat depth is quick and easy to measure, so integrating its measurement into existing national field programs, like Norway's spatially representative nature monitoring or national forest inventory, would be helpful (although not sufficient for regional or local mapping). Municipalities and local actors in Norway are increasingly measuring peat depth and can contribute to a growing data foundation

(Kyrkjeeide et al., 2023). Low-altitude, drone-mounted GPR may prove an efficient approach for collecting many, accurate depth data in a landscape, by combining the advantages of airborne deployment and active sensing (Pelletier et al., 1991; Ruols et al., 2023). All of the above can lay the groundwork for renewed peatland maps.

### 4.3.2 Spatial scale affects model performance and utility

Peat depths typically vary over short distances (e.g. < 1 m), so mapping at 10 m resolution implies that the map will compress

much of the fine-scale variation in depth. For example, large differences in maximum depth can be obscured at 10 m resolution. For the same reason, seemingly small improvements in mean absolute error at 10 m resolution may represent meaningful improvements for map users. Spatial aggregation also means that an uneven distribution of point measurements within a cell can cause its mean depth to be unrepresentative. Some cells had dense and evenly distributed measurements, while others had only a few in one corner. Although we tried to reduce the influence of uneven point coverage – by inversely weighing point

values by their distances to the cell center – it probably contributed to model inaccuracy. Fine-scale variation in peat depth may favor mapping at very fine resolution (1 m) — even if land use planning and carbon accounting do not operate at this grain. Terrain–depth relationships might be stronger in 1 m cells than in our 10 m cells, especially considering the hummock–hollow microtopography of many peatlands (Rydin et al., 1999; Lindsay, 2010).

The scarcity of depth measurements and their short spatial autocorrelation mean that mapping peat depth at broad scales is

not an exercise in spatial interpolation (Hengl et al., 2004). Peat depths in our GPR data showed spatial autocorrelation to a range of 50–100 m, and getting measurements at such fine grain is only realistic for small areas, not across whole landscapes. Therefore, we anticipate that spatially explicit DSM is of limited value in all but the most intensively sampled landscapes (currently nowhere in Norway).

Choosing a spatial extent for DSM can be tricky. A natural starting point is the bounding area around a spatial cluster of depth

measurements. For a given cluster, kNNDM can define how far the boundary can extend beyond the point measurements; if the sample-to-prediction nearest neighbor distribution cannot be simulated by any set of cross-validation folds, then the extent is too expansive (Meyer and Pebesma, 2022; Linnenbrink et al., 2024). However, it is unclear how big (N) any cluster of depths should be, and the tradeoff between multiple small-extent DSM and fewer large-extent DSM needs research. Bohn and Miller (2024) advocate for bottom-up stitching of local DSM, and for peat depth we assert that these should at least stay within

peatland regions (or 'supertopes'), where the composition of mesotopes is similar – e.g. regions dominated by raised bogs versus regions dominated by sloping fens (Moen, 1999; Joosten and Clarke, 2002). Depth varies systematically between bogs and fens (Lindsay, 2016b), between peats formed by terrestrialization versus paludification (Buffam et al., 2010), and probably along other axes of peatland typology. Therefore, DSM is more likely to uncover consistent predictor–depth relationships within peatland regions than across them.





### 4.3.3 Machine learning approaches can build on success


The DSM literature and our results support using flexible machine learning algorithms like RF to predict peat depth. RF avoids negative predictions (c.f. Koganti et al., 2023) and produces good uncertainty estimates (our study, Vaysse and Lagacherie, 2017; Takoutsing and Heuvelink, 2022). As depth data become more abundant, we may move from pixel-based learners to deep learning approaches like convolutional neural networks. Bakkestuen et al. (2023) successfully predicted peatland extent

using convolutional neural networks, and their ability to automatically learn multiscale spatial features would reduce the need to manually engineer these. The success of Multi-Resolution Valley Bottom Flatness as a predictor of peat depth demonstrates that multiscale spatial patterns matter for peat depth and convolutional neural networks are designed to learn such patterns. The kind of relationship described in Buffam et al. (2010), where peat depth in basins related to terrain slope at the basin edge, is also something a convolutional neural network could learn. However, since this approach is data-hungry, we should build

soil knowledge into the DSM where we can (Minasny et al., 2024b). For example, if further research confirms the effect of the marine limit that we found, then it is better to include the marine limit as a predictor than to make the algorithm learn this pattern from elevation independently.

### 4.3.4 Peat extent and depth should be mapped together

Finally, we want to highlight the need for research on peatland extent and peat depth to be better integrated. Since peatland

extent is defined by non-zero peat depth (the specific threshold varies by definition, Minasny et al., 2024a), they are fundamentally linked. The goal, therefore, should be a unified prediction framework for extent and depth. We caution against reducing continuous depth predictions to arbitrary classes (as in Ivanovs et al., 2024; Karjalainen et al., 2025), since classes can be derived from continuous predictions. The distribution of peat depths across full landscapes is zero-inflated, and research is needed to determine whether it is more efficient to parameterize a single model of peat depth (with a larger, generalized dataset)

or to break down the problem into a hurdle model by classifying zero depth and then regressing non-zero depths (with smaller, specialized datasets). Coupling extent and depth will reduce the prevalence of incoherence that we found: deep peat outside the peatland extent and zero depth inside it. A key challenge going forward will be obtaining training data that represents both zero and non-zero components of the depth distribution, since sampling designs often focus on known peatlands.

. R code used in this study is available at https://github.com/julienvollering/DSMdepth. The depth measurements we used are archived at

https://doi.org/10.6073/pasta/6ce440152f693f2156bf5b692a2e7917 and follow data and metadata standards.



**Table A1.** Attributes of the radiometric surveys, as reported in Baranwal et al., (2013) and Ofstad (2015).

|  | Skrimfjella | Ørskogfjellet |
| --- | --- | --- |
| Survey period | 2008-2011 | December 2014–January 2015 |
| Average flight altitude (m) | 75 | 80 |
| Average flight speed ($\mathrm{km\,h^{-1}}$) | 108 | 88 |
| Flight line spacing (m) | 200 | 200 |







**Figure A1.** Calibration of GPR wave velocity at Skrimfjella (a) and Ørskogfjellet (b) by regression of probed depth against wave travel time.





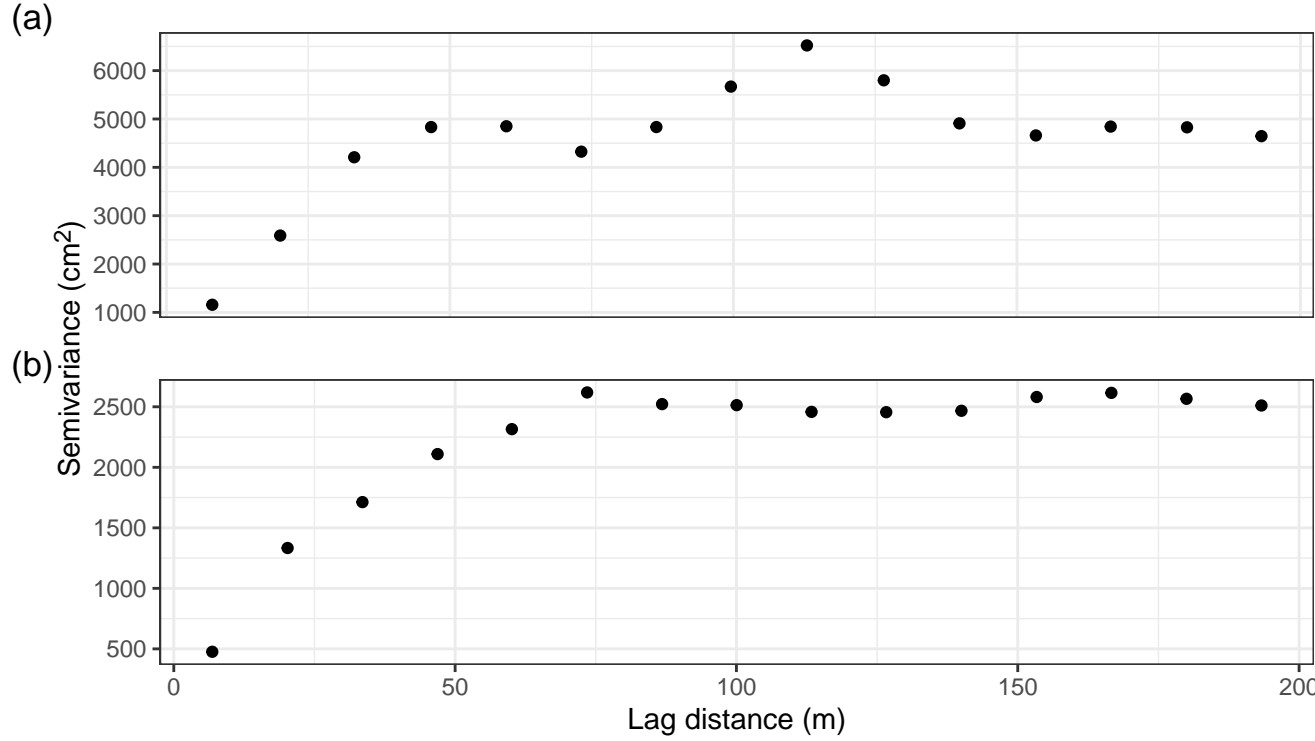

**Figure B1.** Empirical semivariograms of peat depth point measurements at Skrimfjella (a) and Ørskogfjellet (b).

. *Author contributions.* JV: Conceptualization, Investigation, Data curation, Formal analysis, Writing – original draft. NG: Conceptualization, Methodology, Writing - review & editing. MKG: Investigation, Writing - review & editing. KKM: Investigation, Data curation, Writing - review & editing. SDN: Investigation, Writing - review & editing. KR: Conceptualization, Investigation, Writing - review & editing. MS: Investigation, Data curation, Writing - review & editing.

. *Competing interests.* The authors declare that they have no conflict of interest.

. *Disclaimer.* The authors declare that the results, discussions, and interpretations presented in this study are solely their own. The views expressed herein do not necessarily reflect those of their respective institutions or funding agencies.

. *Acknowledgements.* We thank the Norwegian Public Roads Administration for sharing data from ground-penetrating radar surveys. We also thank Vikas Baranwal from the Geological Survey of Norway for helping us access the radiometric data from Skrim. This work




contains data under the following licenses: (1) Creative Commons Attribution 4.0 International, © Kartverket, (2) *Norge digitalt* license, Norwegian Institute of Bioeconomy Research (NIBIO), © Geovekst, and (3) the Norwegian License for Public Data (NLOD), made available by the Geological Survey of Norway. Large language models have been used during the drafting and editing of this manuscript, with author oversight. We maintain full responsibility for the scientific output, as per the European Commission's *Living guidelines on the responsible use of generative AI in research*.





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
