# Peer review of "Terrain is a stronger predictor of peat depth than airborne radiometrics in Norwegian landscapes"

_EGUsphere, 2025_

## Author Response (AR1)

**Authors' Response**

Julien Vollering, on behalf of all co-authors

**Table of contents**

Throughout this document, changes in the manuscript text are indicated with the following formatting:

> Unchanged text.  Added text.

Please disregard missing cross-reference numbers (e.g., 'Figure ??') in the boxes with changed text. These are formatted correctly in the revised manuscript and the tracked changes document.

**Response to Topic Editor**

**Comment TE.1**

> Dear Dr. Vollering and Coauthors, Thank you for your detailed responses to the reviewers' comments. Both reviewers offered positive assessments of your manuscript, recognizing the importance of your study and the inherent challenges in predicting peat depth using a digital soil mapping approach. They also raised several shared concerns, particularly the need for clearer and more formal scientific language, improved manuscript focus, and a more balanced treatment of terrain and radiometric predictors. Your responses to these points appear appropriate, and I encourage you to complete your revisions and submit a revised manuscript for further review.

Thank you very much for this careful and encouraging evaluation. We appreciate the opportunity to submit a revised manuscript for further review, and thank all parties for their time and attention so far.

**Comment TE.2**

> That said, I would like to highlight one important conceptual point raised by Reviewer 1. While it is correct that Random Forest models are invariant to monotonic transformations, such as converting counts/s to concentration, this overlooks the geophysical critique regarding the physical validity of concentration measurements in peat-rich environments. Addressing these physical limitations directly, rather than only statistically, would strengthen the interpretation of your findings. Please ensure that you thoroughly address the issues outlined in Comments 1.5 and 1.6.

Thank you for this advice. Of course the geophysical critique is valid and we do not mean to overlook it. In our revisions we have tried to address these important issues with the attention that they deserve, while also acknowledging that they do not prevent us from treating our research objectives. We have added text in the *Introduction*, *Materials and methods*, and *Discussion* to ensure that readers are made aware of the physical limitations. Please see the sections for Comments 1.5 and 1.6 below, which highlight these specific changes to the text.

**Comment TE.3**

> On a related note, your use of K, U, and Th concentrations as separate predictors differs from the integrated radiometric dose approach used by Gatis et al. (2019), who combined channels to represent total gamma energy. Given the attenuation behavior of gamma rays in wet organic soils, modeling the aggregate energy signal may better capture depth-related variability than using individual concentrations.

I encourage you to clarify this methodological distinction and discuss how it may influence model performance and the interpretation of radiometric predictor utility.

Thank you for drawing our attention to this potential objection. We see that we need to clarify this difference for readers. In short, we use the total count variable provided to use by the surveyors as an integrated radiometric signal. The total count variable is different than the gamma dose rate used by Gatis et al. (2019), but at our study sites they were very highly correlated ($\rho = 0.989, 0.986$). Therefore, our choice to use total count rather than gamma dose rate can only influence model performance and our interpretations of predictor utility negligibly.

Since we now mention this point in the *Materials and methods*, we do not raise it specifically in the *Discussion* – in the interest of brevity.

> **In *Materials and methods***
> We also used the total count variable as provided, rather than calculating a gamma dose rate based on the potassium, thorium, and uranium (as was done in Gatis et al., 2019), because these were highly correlated at both study sites ($\rho = 0.989, 0.986$), and because conversions to dose rates are approximations (IAEA, 2003).

**Comment TE.4**

Your work offers valuable insights into the role of remotely sensed data in peatland mapping. Strengthening the geophysical interpretation within your digital soil mapping framework will enhance the scientific rigor and practical relevance of the manuscript for both soil scientists and geospatial modelers.

Thank you for this endorsement and constructive criticism. We believe that the revisions prompted by the review have strengthened our manuscript for different audiences.

**Response to Referee 1**

https://doi.org/10.5194/egusphere-2025-1046-RC1

**General Comments**

**Comment 1.1**

This article comprehensively addresses the modelling challenges of predicting peat depth from terrain variables. It takes high resolution terrain variables (derived from

LiDAR) and low resolution (comparatively) airborne radiometric data across two individual peat landscapes in Norway and used Random Forest machine learning algorithm to train combinations of these variables to a multitude of peat depth probes and GPR peat depth measurements taken in order to establish the predict power of such variables for peat depth mapping.

Thank you for the time you have kindly taken to review our manuscript.

**Comment 1.2**

Overall, I found the article to be generally well written, with a very comprehensive and detailed description of modelling mechanism, error derivation, and feature choice. It is a very long article, going into great detail in several areas, and below I suggest at least one section/topic that could be removed entirely to reduce length and increase overall readability. I suggest the authors review all sections for conciseness and reduce the article length where possible. The level of detail may mean a reader less familiar with machine learning modelling may find the article hard to follow. While I do find the article to be within the scope of SOIL, I would be concerned that it focuses heavily on the modelling methodology.

Thank you for your positive comments on the article and for your suggestions to improve its readability. We have taken your advice to review the article for conciseness and have made several changes to reduce its length, while retaining the key details of the modelling methodology. In particular, we have removed the sections on peatland extent mapping, which was not central to the main focus of the study, and we have shortened some of the other sections as well. We have moved some of the more detailed descriptions of the methodology to the appendix, to make the main text more accessible and focused.

We do not reproduce all of the removed text here. Please see the tracked changes document to examine these changes.

**Comment 1.3**

Additionally, there were several areas where the language used was quite casual for a scientific article. Several are highlighted under minor considerations below.

Thank you for this advice and the specific examples you provided. We have gone through the manuscript and revised the language to make it more formal and scientific, while aiming to retain clarity and accessibility. A few examples:

> **In *Abstract***
>  Our remote sensing models had better accuracy than the national map

of peat depth, even when we calibrated the national map to the same depth data.
* * *
**In *Introduction***
The  only systematic mapping of peat depth at the national scale in Norway comes from surveys meant to identify arable land.
* * *
**In *Discussion***
Since remotely sensed data are widely available, improvements to soil maps as shown here are  readily achievable at low cost (Minasny et al., 2019).

**Comment 1.4**

> Finally, the main concern noted is the imbalance between the consideration given to radiometric data compared to terrain variables. Considering the title is stating terrain being "better" than radiometrics, and given the emerging understanding of the use of radiometric data in peat land mapping, there is are some fundamental errors in the methods presented, which may be biasing the conclusion alluded to in the title.

Thank you for raising this concern. We address the specific comments on this theme below, and believe that our responses show why we disagree that there are fundamental errors in the methods. For the same reasons outlined below, we do not think that the conclusion the title alludes to is biased.

**Specific Comments**

**Comment 1.5**

> The first concern in the comparison of radiometric data to lidar terrain variables is related to the choice of radiometric variables. The authors opt to use potassium, uranium and thorium ground concentration units alongside the Total Count data from the full energy spectrum. These ground concentration measurements are derived from counts per second measurements on the airplane which are calibrated, usually using pads of known concentrations at a calibration facility. Therefore, concentration of any radioelement is a measurement of the concentration of that element in the top $\sim 60$ cm – 1 m of the soil. However, peat soils are different.

> Being organic, they don't contain the typical geological material that make up soils. Therefore, they act as an attenuative environment to gamma rays. As the potential source of gamma rays in peat areas is blocked and attenuated by the peats, the concentration calibration is no longer physically valid. While these concentration data are indeed provided by the contractors of such surveys, it is now recognized that the counts per second measurement is a more appropriate unit when considering attenuation of gamma rays in peat soils (O'Leary et al, 2022, 2024). In particular considering depth, the deeper the peat, the greater the attenuation of gamma rays. Similarly, the wetter the peat the greater the attenuation of gamma rays. The use of concentration data is not valid for a study in predicting peat depth. I recommend either the authors convert these concentrations to counts per second, or remove all but the Total Count data from their analysis and consider take the next concern into account.

Thank you to the referee for pointing out this concern and providing a thorough explanation of their reasoning. We appreciate the references, which we have consulted.

After careful consideration, we disagree that the radiometrics variables we used are not valid for predicting peat depth. In particular, we assert that converting the concentrations we use to counts per second – as the referee suggests – would produce identical results. That is because the method we use to predict peat depth (Random Forest) is insensitive to any monotonic transformation of the predictors, like the scalar conversion between concentration and counts per second (see Table 10.1 on p.351 in Hastie et al., 2009). The Random Forest algorithm is based on decision trees, which partition the predictor space into regions based on the values of the predictors. As a result, any monotonic transformation of the predictors will not change the partitioning of the predictor space, and therefore will not change the predictions made by the model (see Fig. 9.2 on p.306 and Algorithm 15.1 on p.588 in Hastie et al., 2009). Appendix B3 in Baranwal et al. (2013) and Appendix A2 in Ofstad (2015) confirm that the concentration values we use in our models are simply the counts per second multiplied by a constant conversion factor.

As a separate argument, we also note that Koganti et al. (2023) successfully predicted peat depth with concentration data (K concentration).

To try to avoid confusion on this point, we have added a brief mention of it in the *Introduction*. We also added a short explanation in the *Materials and methods* section that the distinction between counts per second and concentration is not relevant for our analysis and it does not affect the predictive power of the radiometric variables in our models.

> **In *Introduction***
> Another set of predictors related to peat depth are measurements of natural radioactivity from the ground surface. Gamma-ray spectrometry can survey the activity (decay counts per second) from radioactive isotopes in

earth's crust and mineral soils: potassium-40, uranium-238, and thorium-232 and a peat overburden attenuates the radiation intensity at the surface. The degree of attenuation relates to properties of the overburden, especially composition and (Beamish, 2014; Reinhardt and Herrmann, 2019). Although survey measurements are commonly reported as ground concentrations (linearly scaled from decay counts per second), in peatland environments these predictors do not reflect the concentration of radionuclides near the ground surface, but rather the radiation intensity after attenuation by the peat overburden. Deep soil with high water content will attenuate radiation most (Beamish, 2013; Reinhardt and Herrmann, 2019). Thus,  reported ground concentrations can be statistically informative about peat soils, even if not physically correct, with respect to two *scorpan* factors: other soil properties and parent material (McBratney et al., 2003).

**In** *Materials and methods*
Although radiometric data must be in units of counts per second to model attenuation directly (O'Leary et al., 2022), we used the radiometric data as provided to us: in units of concentration for potassium, thorium, and uranium (converted from counts per second by scalar calibration factors). The monotonic transformation between counts per second and concentration has no effect on the tree-based machine learning algorithm that we used to model peat depth (Hastie et al., 2009).

**Comment 1.6**

There is an additional argument missing from within the authors discussion, namely the fact that we never know the initial source strength, or counts, of the gamma rays for a given footprint. The measurement at the airplane is an attenuated version of this initial source. This attenuation is controlled by the attenuation coefficient for a given element and depth, soil moisture, bulk density and porosity (Beamish 2013) of the peat soils. From a purely physics/modelling point of view, this makes the prediction of peat depth an underdetermined problem. Even if the soil moisture, bulk density and porosity was known absolutely, the initial source is never known and so any number of peat depths may result in any given gamma count at the airplane. Additionally, this modelling exercise cannot be performed on Total Count data as this is summed from the entire measurement energy spectrum, which contains multiple element specific attenuation coefficient, meaning the Total Count data is only ever indicative of attenuation variability across a site, with no ability to model anything quantitative. This puts Radiometrics in a natural disadvantage for a quantitative prediction of peat depth. Given the title of this article, I find that the discussion around the radiometrics lacked sufficient detail to make a fair

comparison, which naturally results in such a bias towards terrain variables in such a modelling context. This is therefore not a result per say, but more a perfectly expected outcome. For this concern, I suggest a more comprehensive discussion around the physical limitations of radiometrics in the prediction of peat depth.

Again, we thank the referee for a clear and comprehensive explanation of their concern.

We agree that the physical modelling of peat depth from radiometric data is an underdetermined problem, for the reasons that the referee states. However, our prediction of peat depth is not based on a physical model, but rather on a statistical model that learns the relationship between the predictors (radiometric and terrain variables) and the response variable (peat depth) from the data. This is the same general approach used in previous studies with similar objectives (Keaney et al., 2013; Gatis et al., 2019; Pohjankukka et al., 2025). A mechanistic relationship – in this case the physical attenuation of radiation through peat – is what leads us to expect a statistical relationship in the first place, but the model is otherwise agnostic to the nature and origin of that statistical relationship. That is also why we are able to treat Total Count data the same as the other radiometric variables, even though it can only indicate attenuation variability. Indeed, it is exactly attenuation variability across a site that we expect to be useful for predicting peat depth, and our methods allow the model to decide which of the radiometric variables are most useful and the functional form of the statistical relationship.

It is important to note here that the radiometric and terrain predictors are on equal footing – it is the strength of their statistical relationship with peat depth (including non-linear relationships and interactions with other predictors) that is compared in our analyses and title. Moreover, we assert that there is a clearer mechanistic explanation for a relationship between radiometric variables and peat depth than there is for terrain variables and peat depth. The referee outlines this physical basis for the radiometrics, whereas for terrain variables potential mechanisms are more tenuous, or at least less consistent. For example, terrain slope in peatlands may reflect the topography that underlies the peat layer (e.g. in blanket bogs) or the position along the microtope (e.g. the rand in raised bogs) – neither of which is necessarily strongly related to the thickness of the peat layer. The fact that radiometric measurements integrate across some part of the soil profile is a reason to expect, *a priori*, that these variables might predict peat depth better than measurements that technically only reflect the land surface, like LiDAR and the terrain models derived from it (see for example section 7.2 in Minasny et al., 2019; Reinhardt and Herrmann, 2019; Beamish and White, 2024). We do not think it is correct, therefore, to conclude that there was a bias towards terrain variables in our analysis, or that the results are a perfectly expected outcome.

The referee's comment does expand the list of possible explanations for the poor performance of radiometric variables in our models, and we have added to our discussion based on the comment, as suggested.

In *Introduction*

A complementary approach from soil science is digital soil mapping (DSM). DSM scales up field measurements from a set of locations to a wider area, by relating the measured values to other variables mapped over the area of interest, . This approach has grown in importance with the availability of remotely sensed data and the advancement of  machine learning methods (Minasny et al., 2019; Wadoux et al., 2020).

In *Discussion*

 Another possible reason for the poor predictive value of the radiometric data could be that other physical parameters influencing the amount of intercepted radiation varied too much within sites. Initial source strength, soil moisture, bulk density, and porosity all affect the amount of radiation that reaches the detector (Beamish, 2013; Reinhardt and Herrmann, 2019). Therefore, variation in these parameters could have masked the relationship between peat depth and radiometric data. This makes physical modeling of peat depth from radiometric data an undetermined problem. We chose the Ørskogfjellet site in part because it has a relatively homogeneous bedrock, which should reduce the variation in initial source strength. Soil moisture, bulk density, and porosity, however, are not easily measured across landscape scales and were assumed to be homogeneous. Uneven snow cover and air moisture during the Ørskogfjellet radiometric survey may also have masked the soil signal in these data, as Ofstad (2015) reports large variation in weather conditions. If maps of these other physical parameters at the time of the radiometric survey were available and included in the model, the predictive value of radiometric data might improve, but this is not a practical solution for digital soil mapping of peat depth.

**Comment 1.7**

> The main focus of this article is on the prediction of peat depth. However, the authors include several sections of the possibility of peatland extent mapping. This is not mentioned at all in the abstract, or the introduction in great detail. As this article is already quite long and complex, I suggest the removal of any sections and text related to mapping peat land extent as it is not the focus of the article and only acts to add unnecessary complexity to an already very technical methodology. The authors even mention in Line 535 that their aim was not to map extent. I suggest the removal of all reference to peatland extent prediction and instead focus on the prediction of peat depth. A much shorter reference to the importance of peatland extent knowledge could perhaps be mentioned in the conclusions, but a full analysis and discussion (section 4.1.4) is not appropriate in this article.

Thank you for this helpful suggestion. We initially thought it would be worth including the analysis on extent mapping – especially since our occurrence data at Skrimfjella is a good test set – but based on the comments of both referees, we recognize that it distracts too much from the main focus. We have removed all sections related to mapping peatland extent (including section 4.1.4), to shorten the manuscript and improve its readability. We still discuss briefly the importance of coupling peat depth and extent mapping, since we consider this a research need directly related to the present study, but we have shortened this section as well.
* * *
**In *Discussion***

Finally, we  would like to highlight briefly the need for  peatland extent mapping and peat depth mapping to be better integrated. Since peatland extent is defined by non-zero peat depth (the specific threshold varies by definition, Minasny et al., 2024),  the lateral and vertical dimensions are fundamentally linked. The goal, therefore, should be a unified prediction framework for extent and depth.  Research is needed to determine whether it is  better to parameterize a single model of peat depth  across a full landscape, or to break down the problem into a hurdle model by classifying zero depth and then regressing non-zero depths . Though peatland definitions may encourage reducing continuous depth predictions to arbitrary classes, we caution against this practice (as in Ivanovs et al., 2024; Karjalainen et al., 2025).
* * *
**Technical Corrections**

Line 58- 59. There is no need to include this sentence with an example here, as the next paragraph goes into the necessary detail on Slope. This is an example of how the authors might reduce the overall size of the article.

Thank you. We have removed this sentence and looked to remove similar cases.

Section 2.2 I would suggest moving this opening paragraph to the end of this section as it acts more to sum up how the authors are using the various predictors. It mentions several of the predictors directly, but the are not described until later sections (for example 2.2.1). This would increase the readability of this section.

Thank you for this suggestion. This opening paragraph is intended to give a brief overview of the full suite of predictors before going into a complete description of how each predictor was derived. We think therefore that it is best placed at the start of the section, but we have tried to improve its readability within the structure (including the mentions of specific predictors).

> We created the same suite of peat depth predictors for both sites (25 continuous and 1 categorical; Table **??**).  All continuous predictors were derived either from an airborne radiometric survey or from a DTM. From the radiometric surveys, we simply used the four variables produced by the Geological Survey of Norway: ground concentration of potassium, thorium, uranium, as well as total count. From the DTMs we calculated several land surface parameters, ranging from simple terrain indices to more complex geomorphometric and hydrological variables (Maxwell and Shobe, 2022). The categorical predictor was peat depth class, from a national map dataset. Predictor preparation is described in more detail below.

> Line 179: Remove "also using White Box" as this is obvious.

Thank you.

> Line 258: The authors mention density; however, they do not expand on this. Was this measured in the field, or an operator's observation and subjective interpretation of density?

Thank you. We would have clarified that this was the operator's observation and subjective interpretation, but this section is now removed entirely.

> 371: What is the relevance of the Persons correlation coefficient of 0.7. Was this tested at all? Readers may not be familiar with this so it should be explained a bit more.

Thank you. We have added some more context to explain the 0.7-cutoff:

> Specifically, we eliminated variables from the best performing predictor configuration to obtain a set with no pairwise Pearson correlation coefficient above 0.7 (an arbitrary but conventional threshold for this purpose).

> Table 2: I recommend putting a vertical line between the results for both sites so as to easier distinguish between them.

Thank you. We have reexamined the formatting of the table.

Heading Section 4.1.1 – "but not useless!" is very casual language to be using in a scientific article. This is just one example of this casual language. I suggest the authors review the article for this throughout.

Thank you for this example. We have gone through the manuscript and revised the language to make it more formal and scientific, while aiming to retain clarity and accessibility. We have removed this phrase and similar casual language throughout the manuscript.

Line 464: "low hanging fruit" is also casual and colloquial and may not be understood by all cultures.

Thank you. Removed.

Line 515: "large stocks really are large" – very vague and non-scientific comment. What is large?

Thank you. Clarified:

> This is not unexpected for the right-skewed distributions of peat depth, but it has  implications for potential users of the maps. For example, it makes the maps more suited for "red-lighting" than "green-lighting" peatland conversion  (assuming depth is the  determinative factor). Identifying the deepest peats will require additional field work in candidate areas – where candidate areas could be defined by some upper quantile of predicted depth.

Line 530: remove the question at the start of this section.

Thank you. Removed.

Line 549: This section is the first mention of the radiometric survey parameters. I would recommend moving some of this section to the Methods and Material section as it is useful descriptors of how the data came to be.

Thank you. We have added a very brief summary of the radiometric surveys to the *Materials and methods* section, while retaining the reference to an appendix for the full details.

> The radiometric surveys were  conducted with 75–80 m average flight altitude, 88–108 km h$^{-1}$ average flight speed, and 200 m flight line spacing (Table **??**).

Line 564: Typically, airborne radiometric surveys have strict conditions that they must fly under. One is related to rainfall occurrence and airborne surveys should not happen directly after rainfall. The authors statement with regards to air moisture should be clarified as otherwise the contractor may have been at fault by provide incorrect data, which would have implications for the usage of radiometric data in this area in this study.

Thank you. We have clarified our statement based on the information in the report by the Geological Survey of Norway.

>  Another possible reason for the poor predictive value of the radiometric data could be that other physical parameters influencing the amount of intercepted radiation varied too much within sites. Initial source strength, soil moisture, bulk density, and porosity all affect the amount of radiation that reaches the detector (Beamish, 2013; Reinhardt and Herrmann, 2019). Therefore, variation in these parameters could have masked the relationship between peat depth and radiometric data. This makes physical modeling of peat depth from radiometric data an undetermined problem. We chose the Ørskogfjellet site in part because it has a relatively homogeneous bedrock, which should reduce the variation in initial source strength. Soil moisture, bulk density, and porosity, however, are not easily measured across landscape scales and were assumed to be homogeneous. Uneven snow cover and air moisture during the Ørskogfjellet radiometric survey may also have masked the soil signal in these data, as Ofstad (2015) reports large variation in weather conditions. If maps of these other physical parameters at the time of the radiometric survey were available and included in the model, the predictive value of radiometric data might improve, but this is not a practical solution for digital soil mapping of peat depth.

Line 628 – 629: remove the sentence starting "The rest of this section…." As it is unnecessary.

Thank you. Removed.

Line 634: "luxury" is again a very casual phrasing within a scientific article.

Revised.

Line 664: "tricky" – casual

Revised.

> Finally, there is no definite conclusion to this article, nor a heading stating same. I suggest the authors either add a section at the end and move some text here to highlight the main conclusions as currently the "discussion" section is quite large and probably not appropriate to act as a combined discussion and conclusion section.

Thank you for this helpful recommendation. We have added a *Conclusions* section to summarize the key findings of the study. This summary highlights the main results and their implications, which we hope will help readers more easily see how the study addresses its research aims of quantifying predictive accuracy and identifying key predictors.

We have also used this addition to reduce the length of the *Discussion* section, by reserving some of the more general discussion points for the *Conclusions*.
* * *
**In *Conclusions***

This study demonstrates that digital soil mapping at 10 m resolution can improve upon existing peat depth maps in Norway, though the strength of the relationship between available predictors and peat depth remains limited. Our findings show that terrain-derived variables, particularly elevation and Multi-Resolution Valley Bottom Flatness, provide predictive value for peat depth mapping within peatland extents. In contrast, airborne radiometric data showed little to no predictive value at either of two study sites, possibly because of the large footprint of airborne spectrometers relative to the fine-scale variation in peat depths.

The best models achieved mean absolute errors of 56–60 cm against mean depths of 119–126 cm, and explained approximately one-third of the variation in peat depth across landscapes with aggregate peatland areas of 1.5–15.3 km$^2$. Though field measurements remain necessary for local, detailed assessments of peat depth, digital soil maps at 10 m resolution can provide valuable information for landscape-scale planning and regional carbon assessments. The models' tendency to underpredict the deepest peats has important practical implications, making them more suitable for precautionary screening than comprehensive coverage. As Norway and other nations pursue nature-based climate solutions, these findings highlight both the potential and limitations of remote sensing for peatland carbon mapping.
* * *
**Response to Referee 2**

https://doi.org/10.5194/egusphere-2025-1046-RC2

**General Comments**

**Comment 2.1**

> This article focuses on improving peat depth mapping in two distinct peatland landscapes in Norway using a digital soil mapping approach. The study employed the Random Forest (RF) algorithm to model both peat presence and depth, using high-resolution terrain data, remotely sensed radiometric data, and polygonised peat depth data from an existing map. The RF models were calibrated using field-measured data points collected across both regions, and variable importance was analysed.

Thank you for the time you have kindly taken to review our manuscript.

**Comment 2.2**

> Overall, this research addresses a significant challenge in peat depth mapping. The title is clear and likely to attract attention from readers interested in this topic. However, I have some concerns regarding the writing style. Certain sections use jargon excessively or are phrased in a way that feels less scientific, which may hinder clarity and accessibility for a broader audience.

Thank you for your positive comments on the research and title, and for your suggestions to improve the writing style. We have gone through the manuscript and revised the language to make it more formal and scientific, while aiming to retain clarity and accessibility. We have also tried to reduce jargon where possible, while retaining the necessary technical terms.

**Specific Comments**

**Comment 2.3**

> While the title is clear, I personally find it somewhat overconfident in summarising the results. This is mainly due to the relatively low performance of the RF models and the minimal difference observed between the terrain-only and terrain-plus-radiometric models. To better support the claims, it would be helpful to include a statistical test (e.g., a t-test) to assess whether the performance differences between models are statistically significant. Furthermore, incorporating a stand-alone model that uses only radiometric data as predictors would allow for a more balanced evaluation of the variable groups and help clarify the specific contribution of radiometric inputs in predicting peat depth in Norway.

Thank you for this suggestion. To clarify: the minimal marginal improvement in the terrain-plus-radiometric models compared to the terrain-only models is in fact evidence for the terrain variables being stronger predictors. Nevertheless, you are right that adding a radiometric-only model would provide a clearer comparison of the predictive contribution of radiometric versus terrain data. We now present models with all seven possible combinations of the three variable groups. These expanded results make clearer the predictive gap that we highlight in our title.

We have also added statistical tests to assess the significance of the differences in model performance. The results of these tests are now presented in tables in the appendix. The results show that the differences in model performance are statistically significant in most cases, which supports our claims about the relative importance of terrain variables in predicting peat depth. Specifically, 8 of the 12 direct comparisons between radiometric and terrain variables (Radiometric–Terrain and RadiometricDMK–TerrainDMK configuration pairs * 3 metrics * 2 sites) are statistically significant at the 0.05 level. We now describe these results in the Results section, to better support the claim in the title.

Some of the most important text revisions related to this comment are:
* * *
**In *Materials and methods***
From the cross-validation we quantified mean absolute error (error magnitude, original scale), $R^2$ (explained variation, standardized scale), and Lin's concordance correlation coefficient (error magnitude and explained variation, standardized scale). We formally assessed the effect of predictor configuration on performance metrics using mixed-effects models to account for the cross-validation fold structure (folds as random effects), and testing pairwise differences between configurations.
* * *
**In *Results***
For Skrimfjella, the best predictor configuration was *all predictors*, followed closely by *terrain + radiometric* . The performance gap between the *terrain + DMK* configuration and the *terrain* configuration was similarly small. Compared to the (Fig. **??**). The *terrain* configuration , the outperformed the *terrain + radiometric* configuration improved concordance correlation by 0.04, $R^2$ 0.27 in concordance correlation coefficient ($p < 0.001$; Appendix Table **??**), by 0.04, and 0.16 in $R^2$ ($p = 0.190$; Appendix Table **??**), and by 10 cm in mean absolute error by 1 cm. ($p = 0.038$; Appendix Table **??**). The *terrain + DMK* was a very poor predictor of peat depth even though it was calibrated to measured depths, with a configuration outperformed the *radiometric + DMK* configuration by 0.32 in concordance correlation coefficient of 0.0077 ($p < 0.001$; Appendix Table **??**), by 0.11 in $R^2$ ($p = 0.605$; Appendix Table **??**), and by 11 cm in mean absolute error ($p = 0.011$; Appendix Table **??**).
For Ørskogfjellet, the best predictor configuration was *terrain + DMK*, followed by *terrain* (Fig. **??**). Adding radiometric predictors to these configurations slightly worsened model performance, especially in terms of concordance correlation and $R^2$ . By

 The *terrain* configuration outperformed the *radiometric* configuration by 0.24 in concordance correlation coefficient ($p = 0.068$; Appendix Table **??**), by 0.24 in $R^2$ ($p = 0.004$; Appendix Table **??**), and by 25 cm in mean absolute error ($p = 0.086$; Appendix Table **??**). The *terrain + DMK*  configuration outperformed the *radiometric + DMK* configuration by 0.28 in concordance correlation coefficient  ($p = 0.015$; Appendix Table **??**), by 0.27 in $R^2$ ($p < 0.001$; Appendix Table **??**), and by 28 cm in mean absolute error  ($p = 0.031$; Appendix Table **??**).

**In *Discussion***

Radiometric data had little to no predictive value at  either site (poor performance of *radiometric* configuration), although they did contribute to the best model at Skrimfjella (marginal improvement in *all predictors* configuration compared to *terrain + DMK*). These results contrast with earlier studies that found that radiometrics were useful predictors of peat depth (Keaney et al., 2013; Gatis et al., 2019; Koganti et al., 2023; Pohjankukka et al., 2025).

**Comment 2.4**

Figure 1: While the terrain-look map provides a general view of the study area, the relief of the regions remains unclear. Consider adding a clearer topographic representation (e.g., contour profile) to better illustrate landscape variation. Additionally, it would be helpful to explain the rationale for selecting Skrimfjella, which appears to have relatively limited peat coverage, especially given that the adjacent region seems to contain a larger peatland area.

Thank you for the suggestion to make the relief of the study areas clearer. We have added contour lines the maps of the study areas. We have not added contour profiles between specific points since we think that they may not be as useful for readers as the contour lines, but we are happy to add profiles if the contour lines are not enough.

We have also clarified the rationale for the delineation of the Skrimfjella study area, especially with respect to its limited peat coverage. In short: we were also investigating peatland extent mapping so low coverage was not disqualifying, and accessibility was also an important factor for this field work.

At Skrimfjella we delineated a study area of 34 km$^2$ based on radiometric coverage (limiting to the west) and accessibility ( limiting to the south), as part of a pilot project (Fig. **??**a). In Norway's AR5 national land cover dataset

("areal resources in scale 1:5000", Ahlstrøm et al., 2019), 1.5 km$^2$ (4.5 %) of the study area is classified as 'mire' – defined as areas with mire vegetation and at least 30 cm of peat depth. Relatively sparse peatland cover did not disqualify the area for our purposes, since we were also interested in peatland extent mapping in the pilot project.

**Comment 2.5**

> It is unclear how the peat depth data are distributed, both statistically and spatially. While the sampling design is described in the text, I recommend including a figure showing the spatial distribution of the data points to help readers better understand the coverage and representativeness of the dataset. Additionally, a basic statistical summary (e.g., mean, median, range, standard deviation) of the peat depth values would be beneficial to provide context and a clearer picture of the conditions being modelled.

Thank you for this suggestion. We have added a figure showing the statistical and spatial distribution of the peat depths in the Results section. We have taken care to use the same map format as in the figure that presents land cover in the study areas, which we hope will help readers see the spatial representativeness of the dataset.

**Comment 2.6**

> Since the radiometric data are originally at 50 m resolution and the terrain data are at 1 m resolution, the method used to resample these datasets to a common 10 m resolution could influence model performance. In particular, the use of cubic spline resampling for the radiometric data may affect how well its spatial variability is represented, especially when compared to the aggregated 10 m topographic data. I would be interested to hear your thoughts on how this resampling approach might have impacted the results.

We appreciate this attention to the potential impact of radiometric resampling methodology on our results. To address this concern, we added to our code a sensitivity analysis comparing cubic spline and bilinear resampling methods for downscaling the 50 m radiometric data. Using field measurement locations from both study sites, we extracted radiometric values resampled by both methods and calculated correlations for each radiometric variable (K, Th, U, TC). The correlations between cubic spline and bilinear resampling methods were very high, with the lowest correlation being 0.995. This near-perfect correlation demonstrates that the choice of resampling method has negligible impact on the spatial representation of radiometric variability at our target resolution.

The consistently high correlations across all radiometric variables indicate that our conclusions about radiometric predictive performance would remain unchanged regardless of resampling method. Since we now mention this fact in the *Materials and methods*, we do not raise this point specifically in the *Discussion* – in the interest of brevity.

> **In *Materials and methods***
>
> The Geological Survey of Norway conducted and processed radiometric surveys over our study areas, as reported in Baranwal et al. (2013) and Ofstad (2015). They provided us for each site four variables at 50 m resolution, which we downscaled to 10 m resolution by cubic spline resampling, using the *terra* package (v1.7) in R. Sensitivity analysis showed that Pearson correlations between cubic spline and bilinear resampling methods exceeded 0.995 for all radiometric variables, so we are confident that the choice of resampling method did not affect our results.

**Comment 2.7**

> I found the section on sampling design and peat depth measurement to be overly detailed for the main text. While this information is valuable, it may be more appropriate to move some of it to the appendix or supplementary material, as it is less central to the analysis and interpretation of results. This would help improve the flow and focus of the main manuscript.

Thank you for this helpful suggestion. We agree that this an acceptable way to reduce the length of the main text and improve its readability, especially with the added figure showing the spatial distribution of depth measurements (in response to comment 2.5). We have moved the bulk of the sections on *Peat depth sample selection* and *Depth measurements* to the appendix, while retaining brief summaries of these in the main text. This solution seems to be allowed under the journal's guidelines for manuscript composition.

**Comment 2.8**

> In the model interpretation section, it appears that three different methods were used to assess variable importance: FIRM, permutation importance, and Shapley values. However, the implications of using these different approaches are not clearly discussed. Each method captures different aspects of variable influence and may lead to different interpretations. Could you clarify how the results from these methods align or differ, and what their respective implications are for understanding the drivers of peat depth in your study?

Thank you for this suggestion. We have added text to the *Materials and methods* and *Results* sections to clarify how the three variable importance methods relate to each other. We have

also added Spearman rank correlation coefficients to our figure on variable importance, to highlight the similarities and differences. In short, we find that Shapley and permutation importance are very similar for our application, while FIRM values capture a different aspect of variable importance.

Since the different methods are broadly congruent and their differences do not affect our interpretation much – and in the interest of brevity – we do not go into this point in our *Discussion.*
* * *
**In *Materials and methods***

We calculated variable importance with the *vip* R package (v0.4), by three different methods: *FIRM*, *permutation*, and *Shapley* (Greenwell and Boehmke, 2020). *FIRM* values measure the flatness of the partial dependence plot, *permutation* values measure the decrease in model performance when the predictor is permuted, and *Shapley* values are aggregated from local, game-theoretical measures of variable importance (Greenwell and Boehmke, 2020). Since *FIRM* reflects the flatness of the partial dependence plot, it captures functional complexity rather than overall predictive impact. *Permutation* values were obtained from ten iterations, with root mean square error as the performance measure.
* * *
**In *Results***

For the purpose of model interpretation, the *all predictors* configuration for Skrimfjella was reduced from 27 variables to 11 non-collinear variables, by removing one of the variables in each highly-correlated pair. Similarly, the *terrain + DMK* configuration for Ørskogfjellet was reduced from 23 variables to 11 non-collinear variables. The *permutation* and *Shapley* methods of variable importance showed high rank correlation at both sites, while the *FIRM* method ranked variable importance differently (Fig. **??**).

**Variable importance**

At both sites, elevation and Multi-Resolution Valley Bottom Flatness were important predictors (Fig. **??**). At Skrimfjella these two predictors were of similar importance, while at Ørskogfjellet elevation  had higher predictive impact (*permutation* and *Shapley*) but lower functional complexity (*FIRM*) than Multi-Resolution Valley Bottom Flatness. DMK was also important – the  *shallow* class in particular – but only at Ørskogfjellet. Some realizations of the hydrological predictors Topographic Wetness Index and Depth-to-Water showed considerable importance, while others showed little – with no clear consistency between sites.  At both sites, the most important  hydrological predictors were more important than the simple terrain indices slope, Terrain Ruggedness Index, Topographic Position Index, and
* * *
roughness.  At Skrimfjella, the radiometric predictor uranium ground concentration – which was highly correlated with all other radiometric variables – showed moderate importance .

**Comment 2.9**

> I recommend adding a dedicated section to summarise the key results and findings. This would help readers clearly see how the study addresses its original research aims and questions. A concise summary at the end of the results or discussion section would also improve the overall structure and coherence of the paper.

Thank you for this helpful recommendation. We have added a *Conclusions* section to summarize the key findings of the study. This summary highlights the main results and their implications, which we hope will help readers more easily see how the study addresses its research aims of quantifying predictive accuracy and identifying key predictors.

> **In** *Conclusions*
>
> This study demonstrates that digital soil mapping at 10 m resolution can improve upon existing peat depth maps in Norway, though the strength of the relationship between available predictors and peat depth remains limited. Our findings show that terrain-derived variables, particularly elevation and Multi-Resolution Valley Bottom Flatness, provide predictive value for peat depth mapping within peatland extents. In contrast, airborne radiometric data showed little to no predictive value at either of two study sites, possibly because of the large footprint of airborne spectrometers relative to the fine-scale variation in peat depths.
>
> The best models achieved mean absolute errors of 56–60 cm against mean depths of 119–126 cm, and explained approximately one-third of the variation in peat depth across landscapes with aggregate peatland areas of 1.5–15.3 km$^2$. Though field measurements remain necessary for local, detailed assessments of peat depth, digital soil maps at 10 m resolution can provide valuable information for landscape-scale planning and regional carbon assessments. The models' tendency to underpredict the deepest peats has important practical implications, making them more suitable for precautionary screening than comprehensive coverage. As Norway and other nations pursue nature-based climate solutions, these findings highlight both the potential and limitations of remote sensing for peatland carbon mapping.

**Technical Corrections**

> Table 2: I think you need to move this table somewhere after it is mentioned in the main text. At first, I was quite confused with this table. What does that

We have moved the table after its first mention and adjusted the table caption to clarify that the percentages represent the proportion of the total number of 10 m cells across the stratifications. We have also added standard deviations to the mean depths, in accordance with comment 2.5.

> Line 394-397: I don't quite get the point of this section. To my understanding, you compared between the quartile predictions and the observed data to see the prediction interval coverage probability. If so, I think it would be better to plot this prediction interval together with the observed data in scatter plot, like in Figure 3.

Thank you for identifying this confusion. We have revised the text to clarify that we are comparing the quartile predictions to the observed data, to follow best practices in digital soil mapping. We have also added the prediction intervals to Figure 3 (Figure 4 in the revised manuscript), and moved the text description to where we also present the figure. We think that this makes this particular result clearer to readers.

> The best models at both sites overpredicted shallow peats and strongly underpredicted very deep peats (Fig. **??**). The mean error (bias) of these models was 10 cm at Skrimfjella and -4 cm at Ørskogfjellet. Although the prediction intervals from the quantile regression forests were wide, they were well calibrated. At Skrimfjella, the prediction interval coverage probability was 92 %, and at Ørskogfjellet it was 84 % (both compared to the target value of 90 %). Observations outside of the prediction intervals showed no obvious spatial pattern.

> Line 413-414: which curve? The information inside the parentheses is not clear to me.

Thank you. We would have revised this for clarity, but this entire section has been removed.

> Figure 4: is the null derived from the calculation between observations and assumptions of 30 cm peat depth? Where is the standard error come from?

We have removed this figure along with all sections about peat extent mapping. The null did reflect the error between observations and assuming 30 cm depth. The standard errors on the other error estimates were from the cross-validation folds.

> Line 464: "low hanging fruit" ?

Revised.

> Line 515: This sentence is unclear to me.

Revised. The sentence was intended to reiterate what came before, so we removed it for clarity conciseness.

> Line 531: "0% in prediction" what does this sentence mean?

Removed along with all sections about peat extent (occurrence) mapping.

> Line 614: What are the 'both indicators variables' referring to?

Revised for clarity.

**Response to editorial support team**

**Comment ES.1**

> Please add the section headlines "Appendix A" (in front of Table A1) and "Appendix B" (in front of Figure B1).

Thank you. Revised.

**References**

[revised manuscript text omitted]

---

## Author Response (AR2)

**Authors' response #2**

Julien Vollering, on behalf of all co-authors

We thank the topic editor for accepting our manuscript. We also thank the the reviewers again for their efforts.

We are not aware of any comments on our revised submission, or specific requested changes. The only change we have made in the text with respect to the revised submission is that we have included an in-text citation to our archived data, as requested in the email from the editorial support team. We have also added a new affiliation for one author.